# Disentanglement Beyond Static vs. Dynamic: A Benchmark and Evaluation Framework for Multi-Factor Sequential Representations

**Tal Barami**[*] **Nimrod Berman**[*] **Ilan Naiman**[*] **Amos H. Hason** **Rotem Ezra** **Omri Azencot**
Faculty of Computer and Information Science
Ben-Gurion University of the Negev
`{baramit, bermann, naimani, hasona, rotemez}@post.bgu.ac.il`
`azencot@bgu.ac.il`

## Abstract

Learning disentangled representations in sequential data is a key goal in deep learning, with broad applications in vision, audio, and time series. While real-world data involves multiple interacting semantic factors over time, prior work has mostly focused on simpler two-factor static and dynamic settings, primarily because such settings make data collection easier, thereby overlooking the inherently multi-factor nature of real-world data. We introduce the first standardized benchmark for evaluating multi-factor sequential disentanglement across six diverse datasets spanning video, audio, and time series. Our benchmark includes modular tools for dataset integration, model development, and evaluation metrics tailored to multi-factor analysis. We additionally propose a post-hoc Latent Exploration Stage to automatically align latent dimensions with semantic factors, and introduce a Koopman-inspired model that achieves state-of-the-art results. Moreover, we show that Vision-Language Models can automate dataset annotation and serve as zero-shot disentanglement evaluators, removing the need for manual labels and human intervention. Together, these contributions provide a robust and scalable foundation for advancing multi-factor sequential disentanglement. Our code is available on GitHub, and the datasets and trained models are available on Hugging Face.

## 1 Introduction

Learning disentangled representations has become a core research focus in deep learning [3], with applications across vision, audio, time series, and language domains [25, 40, 33, 17]. The goal is to map data into sub-representations that capture distinct semantic factors [57]. Disentangled representations boost model performance, fairness, controllability, and interpretability [37, 15, 17]. Due to scarce labeled data, much work has focused on *unsupervised* disentanglement learning [36]. A key subfield is *sequential disentanglement* [26, 34], which traditionally aims to learn two representations from sequential data: dynamic features that evolve over time and static features that remain invariant. For example, in the case of a smiling person, their identity remains static while the facial expression changes dynamically.

Two key challenges in sequential disentanglement remain open. The first is *multi-factor disentanglement*: rather than treating static and dynamic aspects as single units, it is crucial to further decompose each into multiple meaningful sub-factors [8, 6]. For instance, a person's identity (static) may be further decomposed into age and sex, while dynamic facial patterns might include smile intensity, head

---

[*]Equal contribution

movement, and motion speed. The second challenge is achieving *modality-agnostic disentanglement*. Prior work in sequential representation learning [34, 2, 6, 44] has aimed to develop general-purpose methods that are agnostic to the underlying data modality. This offers a key advantage: models can be applied across different data types without requiring significant architectural or algorithmic adaptations. Addressing both challenges would enable finer control, deeper analysis, and broader applicability of sequential models. Despite its potential, multi-factor and modality-agnostic sequential disentanglement remains relatively underexplored.

In this work, we tackle three critical barriers to progress in multi-factor disentanglement. (1) The field lacks standardized infrastructure: unlike two-factor disentanglement, which benefits from established benchmarks [64, 2, 44], multi-factor research suffers from fragmented datasets, inconsistent evaluation protocols, and limited public code [59, 8], often relying on synthetic data with limited real-world relevance. (2) Current protocols depend on fully labeled datasets, which are scarce in practice. Even when labels are available, additional training of dataset-specific classifiers is often required, demanding domain expertise and significant resources. (3) Evaluation is labor-intensive: even if labels are available, mapping semantic factors to latent variables typically requires manual inspection, especially in non-compact settings where a single factor maps to multiple latents [6].

To overcome the first challenge, we compile a diverse benchmark spanning six datasets across multiple modalities. This includes two existing video datasets, two new synthetic datasets generated using established tools, a novel audio dataset, and a real-world time series dataset with extracted feature labels. All datasets are provided in a unified format to eliminate inconsistencies and alignment issues. To support fair comparison, we adapt and standardize implementations of existing disentanglement methods from prior work, and additionally propose four new baselines, including one novel method that outperforms previous state-of-the-art results. For evaluation, we present ten metrics. We refine four existing sequential disentanglement metrics, adopt three image-based disentanglement metrics for sequential setup, and augment three new disentanglement-consistency metrics for the video domain. Overall, these contributions form a comprehensive development-evaluation framework for fair, thorough evaluation across datasets, methods, and metrics.

To address the second challenge, we propose using *Vision-Language Models (VLMs) as taggers and judges* [61] for video-based sequential disentanglement. Inspired by LLM-based judges [62, 21] and recent advances in VLMs [12, 31], we leverage VLMs to perform two key tasks for datasets lacking ground truth labels: (a) *Tagger* – the unsupervised discovery of semantic factors of variation (e.g., hair color, lighting) and the corresponding identification of their label spaces (e.g., blonde, gray, brown); and (b) *Judge* – a zero-shot evaluator for diverse video modalities that can replace domain-specific classifiers in existing evaluation pipelines. Together, these components pave the way for almost-fully automated workflows and facilitate the use of real-world video datasets. While our use of VLMs represents an early step in this direction, we demonstrate their effectiveness on both synthetic and real-world benchmarks and provide a user-friendly API for broader adoption.

As for the third challenge, we introduce a dedicated *latent exploration stage*. Traditionally, evaluating disentanglement methods required manually inspecting and matching latent variables to semantic factors, even when ground truth labels were available. This process was not only tedious but also inconsistent and difficult to scale. Our post-training phase automates this step: given a trained model, its dataset, and corresponding labels, it identifies which latent variables correspond to which semantic attributes. This removes the bottleneck of manual mapping and enables rapid, reproducible benchmarking. In addition, we propose two model-agnostic heuristic strategies that are robust, plug-and-play, and easily extensible. Empirical results show that this automated approach not only reduces human effort but can also surpass manual matching in complex scenarios.

We conduct an extensive quantitative comparison across **6** methods, **6** diverse datasets, and up to **10** evaluation metrics, offering a comprehensive overview of the current state in multi-factor sequential disentanglement. To expose the limitations of existing methods, we analyze a representative failure case on high-quality video data. We also validate the accuracy and reliability of our VLM-based evaluation pipeline on synthetic and real-world datasets. Finally, we outline promising future directions and key open challenges. **Overall**, our benchmark, including the automation tools and new baselines, establishes a scalable and principled framework for the development, evaluation, and comparison of disentanglement methods, aiming to catalyze the next wave of progress in this field.

## 2 Related work

**Non-sequential disentanglement and benchmarking.**  Disentanglement has long been a central goal in unsupervised learning and generative modeling [3]. Early methods introduced inductive biases to encourage disentanglement without relying on explicit supervision [23, 14, 27]. In the vision domain, benchmarks such as dSprites [24], 3D Shapes [27], and the extensive analysis in [36] have been instrumental in advancing the field by providing datasets with ground truth factors of variation that enabled rigorous evaluation and catalyzed research on disentanglement methods. However, these benchmarks focus exclusively on static images and fail to capture the rich temporal dynamics inherent in sequential data. To address this limitation, we adapt existing datasets and develop new generators capable of producing video sequences with controlled, time-varying factors, thus extending disentanglement benchmarking to the multi-factor sequential domain. We hope our benchmark will similarly serve as a catalyst for progress in multi-factor sequential disentanglement research.

**Two-factor sequential disentanglement and benchmarking.**  Separating static and dynamic factors in sequential data has been extensively studied in the video and speech domains [26, 34]. Early models, such as FHVAE [26], introduced explicit architectural mechanisms to disentangle these two components. Subsequent approaches have incorporated theoretical guarantees and architectural biases to further improve sequential disentanglement quality [64, 2, 22, 44, 5, 65]. In parallel, evaluation benchmarks have expanded to cover video, audio, and time series data, with datasets such as Sprites [34], MUG [1], TIMIT [66], and PhysioNet [19]. However, multi-factor evaluation remains challenging: most datasets annotate only coarse static and dynamic attributes (e.g., identity or expression), overlooking the finer-grained factors that compose them, such as hair color or sex within identity. As a result, while two-factor evaluation protocols are now relatively well-established, extending them to support multi-factor sequential disentanglement across modalities remains a significant open problem.

**Multi-factor sequential disentanglement and benchmarking.**  These approaches seek to disentangle multiple static and dynamic factors simultaneously, moving beyond the traditional binary separation. Notably, [8] introduced a framework for disentangling multiple latent attributes in videos, though their work was limited by reproducibility and dataset availability. Similarly, [59] proposed a sequence-based model but faced similar challenges. More recently, [6] advanced the field by releasing a standardized evaluation protocol and public codebase. Related but distinct are recent works on symmetry learning and Meta-Sequential Prediction (MSP) [41], and its extensions via Neural Fourier Transform (NFT) [30] and Neural Isometries [39], which approach disentanglement through the lens of latent linear operators and equivariant structures. Despite these improvements, evaluation still relies heavily on manual matching between latent variables and semantic factors, limiting scalability and large-scale comparisons. Moreover, the diversity and availability of multi-factor sequential datasets lag behind those developed for two-factor disentanglement. As a result, the landscape remains fragmented, with custom datasets and inconsistent evaluation criteria. No prior work has addressed the need for automated evaluation at the multi-factor level. In this work, we aim to bridge these gaps by introducing a unified, modality-diverse, and scalable benchmark, along with tools for automated evaluation and broader dataset coverage.

**VLMs as taggers and judges.**  The use of LLMs as automated judges has gained significant traction, with several studies [62, 35, 21] demonstrating their reliability in evaluating tasks like summarization, dialogue generation, and translation. Building on this success, VLMs have been adapted for judging multimodal tasks [12], including image captioning, visual question answering (VQA), and referring expression comprehension. Chen et al. [12] showed that Multimodal LLMs (MLLMs) achieve strong alignment with human judgments in VQA settings. Beyond evaluation, VLMs are increasingly used for automatic tagging and annotation: previous work leveraged VLMs to construct web-scale datasets, and incorporated VLM-generated labels into benchmark datasets [63]. Building on these advances, we leverage VLMs both as taggers for discovering semantic factors and as judges for unsupervised model evaluation, enabling fully automated pipelines for dataset annotation and evaluation in multi-factor sequential disentanglement tasks.

## 3 The Multi-factor Sequential Disentanglement (MSD) benchmark

In what follows, we define the problem setting (Sec. 3.1) and outline the benchmark's main components—datasets (Sec. 3.2), methods (Sec. 3.3), and metrics (Sec. 3.4). The benchmark is modular, with standardized interfaces allowing users to add datasets, implement methods, or extend metrics

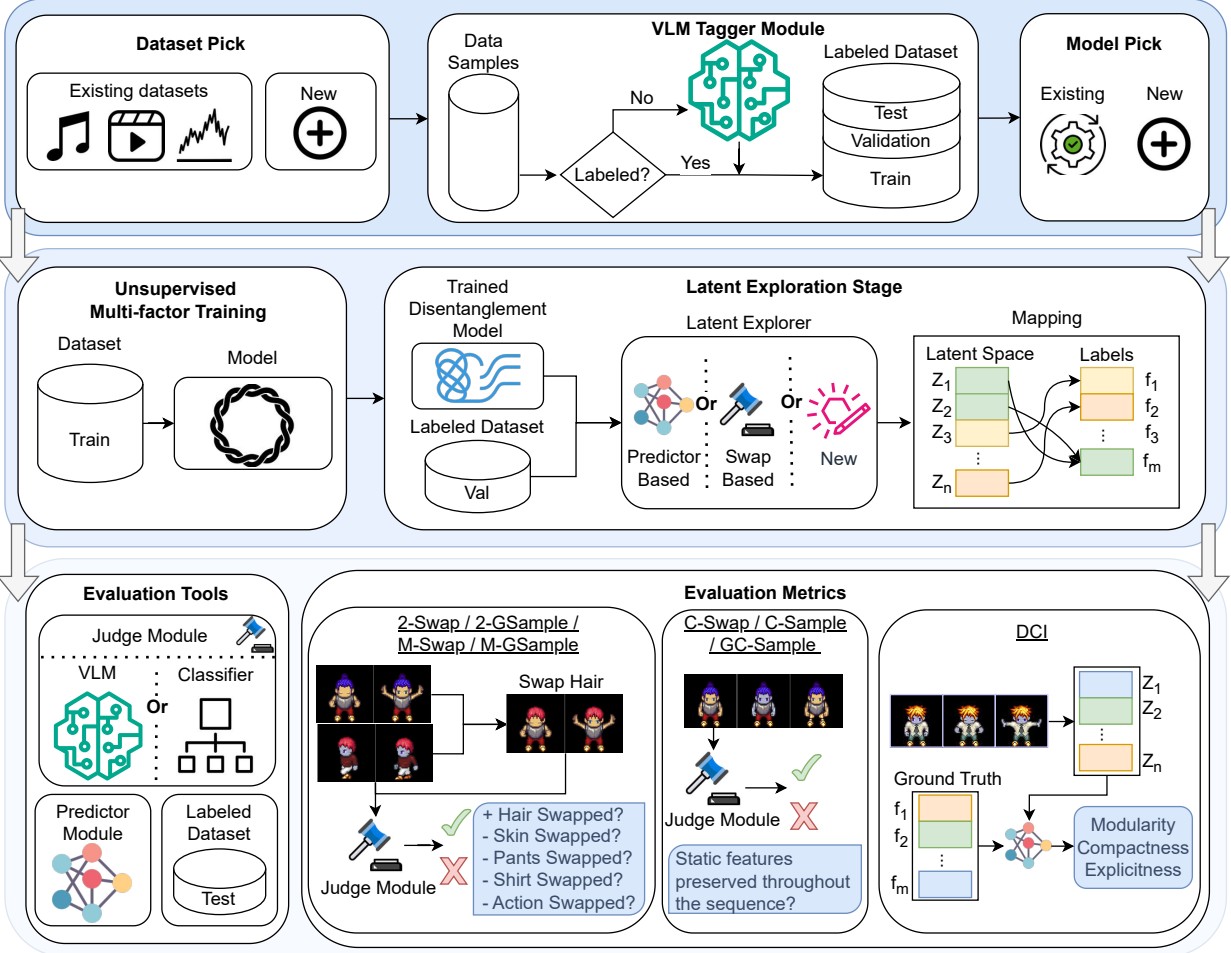

Figure 1: A visual summarization of our benchmark codebase and user flow. A full explanation of all the details of the flow can be found in App. A.

seamlessly. We then introduce the Latent Exploration Stage (Sec. 3.5), the Koopman-based multi-factor disentanglement method (Sec. 3.6), and the VLM-based judge (Sec. 3.7). Fig. 1 summarizes the framework.

## 3.1 Multi-factor sequential disentanglement

Let $X$ be a sequential dataset where each sample $x \in X$ is a sequence of length $T$ where each element $x_t$ lies in $\mathbb{R}^o$. For example, in a multivariate time series $o$ is simply the number of features per time step; in video $o = c \times h \times w$ denotes a structured image tensor. We assume variation in $X$ is governed by a finite set of semantic factors $F = \{f_1, \ldots, f_k\}$ [3], each associated with a discrete label space $\mathcal{Y}_{f_i} = \{y_{f_i}^1, \ldots, y_{f_i}^n\}$. While $k$ and $n$ may be unbounded in theory, we assume both are finite for practical purposes. A disentanglement model $\mathbf{M}$ learns latent representations $z \in \mathbb{R}^l$, aiming to assign each factor $f_i$ to a subset of coordinates $z_{J_i}$, where $J_i \subseteq \{1, \ldots, l\}$. We denote the full mapping as $J_{\text{map}} = \{J_1, \ldots, J_k\}$, ideally disjoint to ensure each latent subset encodes only one factor. To handle sequential data, we partition $F$ into disjoint static and dynamic subsets: $F = S_F \cup D_F$, with $S_F \cap D_F = \emptyset$. Static factors (e.g., identity, sex) remain constant across time, while dynamic factors (e.g., pose, expression) vary. The model must disentangle $z$ such that each subset of features $z_{J_i}$ exclusively controls its associated factor $f_i$, faithfully reflecting its temporal behavior—static or dynamic.

## 3.2 Datasets

We curate a collection of six diverse datasets spanning three modalities, all of which adhere to the formal definition outlined above. For the video modality, we include Sprites [34] and create sequential

variants from the established 3D Shapes [27] and dSprites [24] (two variants: dSprites-Static and dSprites-Dynamic). For audio, we create the realistic-sounding dMelodies-WAV dataset, which builds upon dMelodies [48], while for time series, we adapt and label the real-world BMS Air Quality dataset [13]. In addition to the datasets we release, a major focus of our benchmark design is to make integrating new datasets as seamless as possible. We encourage the community to contribute datasets in our standardized format, fostering a collective effort. Datasets without explicit $F$ and $\mathcal{Y}$ information can also be included; however, in such cases, latent exploration and evaluation functionalities will be disabled. Further implementation and annotation details are provided in App. B.2, and a summary table (Tab. 4) highlights the key characteristics of each dataset.

### 3.3 Methods

To fully-participate in our benchmark, methods must support, through our API, a standardized set of functionalities, such as ingestion of multi-factor sequential datasets and integration with our latent exploration and evaluation stages. Specifically, each method must be able to: (1) provide encoding and decoding functions to and from its latent space; (2) output a fixed-length representation (e.g., a 1D vector) for each input sequence, obtained by flattening or aggregating the full latent sequence; (3) swap channels of two latent representations; and optionally be able to: (4) sample new data points from its latent space. As part of building the benchmark, we made a comprehensive effort to implement and adapt a broad set of available multi-factor sequential disentanglement methods. Some were integrated directly, while others required substantial refactoring, and for methods lacking official implementations, we re-implemented them from scratch. Despite best efforts, a few methods could not be reliably reproduced. To further enrich the benchmark, we introduced several new baselines, all included in our released codebase. Thanks to the modular framework, methods are fully decoupled, allowing new additions with minimal effort. The benchmark includes the following models: Sequential VAE [28]; Sequential $\beta$-VAE [23]; Sequential Sparse-AE [47]; Multi-disentangled-features Gaussian Processes VAE (MGP-VAE) [8], which leverages Gaussian processes to model static and dynamic features; Structured Koopman Disentanglement (SKD) [6], which is based on spectral loss and Koopman operator; and our improved Single Static Mode SKD (SSM-SKD), which we present in Sec. 3.6, enhancing latent inference and static decomposition. We also attempted to incorporate methods from [25, 59]; however, [59] lacks a public implementation and could not be reproduced, and while [25] provides code, we were unable to adapt it reliably to our framework. The benchmark is designed for continuous expansion of method coverage. A full summary of implementation status is provided in Tab. 5, and further architectural details appear in App. C.

### 3.4 Metrics

We propose ten metrics that capture complementary aspects of multi-factor disentanglement. They are grouped into three categories, each assessing a distinct property of disentangled sequential representations. Some metrics are modality-specific, such as those evaluating consistency in video data, while others are modality-agnostic and applicable across domains.

**Factorial swap and sample metrics.** The first group evaluates whether latent subspaces exert precise and selective control over semantic attributes in the output. (1) *2-Swap* and (2) *2-GSample*, derived from existing benchmarks [44, 2], test whether models can correctly separate static and dynamic factors in two-factor setups by swapping or sampling latent components and verifying outcomes with classifiers or vision–language models (VLMs). (3) *M-Swap* and (4) *M-GSample* extend this evaluation to the multi-factor setting, following [6]. They test whether a model can manipulate or preserve individual factors selectively. Each of these four metrics generates numerous values that complicate model comparison; thus, we introduce a weighted mean score to summarize these values into a single value that balances preservation and controlled variation. These metrics are particularly well-suited for video and audio datasets, where factor-level interventions produce semantically interpretable results. In contrast, they are excluded from time series datasets, where such interventions are difficult to interpret (e.g., swapping the "month" in weather records).

**Disentanglement-Completeness-Informativeness (DCI) metrics.** Metrics (5)–(7) extend the DCI framework [18], adapted to sequential and multi-factor settings. (5) *DCI-M (Modularity)* measures whether each group of latent dimensions affects only a single factor. (6) *DCI-C (Compactness)* evaluates whether each factor is controlled by a small subset of latents. (7) *DCI-E (Explicitness)* quantifies how well latent representations retain predictive information about the factors. These

metrics are modality-agnostic and are applied consistently across all benchmark datasets. We adapt these metrics for sequential data by flattening latent trajectories to form set-level representations and compute disentanglement scores at the factor level. This adaptation draws inspiration from established toolkits such as `disent` [38] and `disentanglement_lib` [36].

**Consistency metrics.** The final group comprises video-specific metrics (8–10), which evaluate the temporal stability of static factors across generated or manipulated sequences, inspired by recent video consistency metrics [53]. (8) *C-Swap* measures whether static attributes remain consistent when swapped between two video sequences. (9) *C-Sample* checks whether sampling static factors produces temporally coherent outputs. (10) *GC-Sample* evaluates global consistency by verifying whether static attributes are preserved across entire generated sequences. These metrics focus on temporal coherence and are applied exclusively to video datasets.

**Implementation and scope.** The DCI metrics provide a domain-agnostic foundation, while the swap, sample, and consistency metrics extend disentanglement evaluation into sequential and modality-specific regimes. Intervention-based metrics (2-/M-Swap, 2-/M-GSample) are used only where factor manipulations are semantically meaningful, while consistency metrics are restricted to visual modalities. A detailed discussion of all metrics is provided in App. D.

### 3.5 Latent Exploration Stage (LES)

Evaluating unsupervised disentanglement is inherently challenging [51], particularly for our setup that does not include the *compactness assumption*, i.e., semantic factors like age or hair color may span multiple latent dimensions [11]. This complicates interpretation and often requires extensive human intervention to identify the axes of variation. Even when models are trained successfully, manually assigning latent dimensions to semantic factors is not only time-consuming but can also degrade performance. To address this, we introduce the *LES*: a post-hoc procedure that, given a dataset, its labels, and a trained model, automatically maps latent subspaces to semantic factors. Demonstrating its effectiveness, applying LES to SKD [6] not only reduces the need for human effort but also improves results on multi-factor tasks, for instance (M-Swap: $0.69 \rightarrow 0.70$, M-GSample: $0.67 \rightarrow 0.71$), underscoring its value for scalable and reliable evaluation.

The benchmark includes two complementary LES strategies: *predictor-based* and *swap-based*. The predictor-based method trains a supervised classifier per factor to predict ground truth labels from the latent code; feature importances then reveal which dimensions are most relevant. This approach is fast and effective when labels are available, but may degrade under noisy supervision. In contrast, the swap-based method performs latent interventions: it swaps a subset of latent dimensions between samples, decodes the results, and uses a pretrained judge to identify which factor has changed. Repeated trials reveal latent–factor associations. Although more computationally intensive, this method captures subtle and combinatorial relationships that predictor-based strategy may miss. Both approaches are modular and extensible, and users can add new exploration methods with minimal effort. Additional details on LES are provided in App. F.

### 3.6 Single Static Mode Structured Koopman Disentanglement (SSM-SKD)

**SKD.** Our method builds on the SKD framework introduced in [6]. SKD is grounded in Koopman operator theory [29, 9], which states that the dynamics of a nonlinear system can be represented linearly in a lifted (potentially infinite-dimensional) space via a linear operator known as the Koopman operator. SKD leverages this idea by approximating the Koopman operator with a finite-dimensional matrix, allowing nonlinear temporal dynamics to be modeled as linear transformations in a learned latent space. This is realized using Koopman AEs, which jointly learn the latent encoding and a Koopman matrix estimated per batch [55, 46, 43]. To encourage disentanglement, SKD enforces a spectral structure on the Koopman matrix: some eigenvalues are fixed at one, while others are constrained to lie strictly within the unit circle. This spectral constraint decomposes the latent space into time-invariant and time-variant components via projection onto the corresponding eigenspaces.

**SSM-SKD.** While SKD separates static and dynamic components, it does not guarantee disentanglement among multiple, e.g., static, factors, as each Koopman eigenvector may entangle several features due to a lack of orthogonality. To mitigate this, we propose encoding static information in a *single* eigenvector, with individual factors represented across its orthogonal coordinates. We focus on static disentanglement and thus designate one eigenvector for static content, treating all others as dynamic. However, forming a single static vector at the batch level is insufficient—one vector lacks the capacity to capture diverse static information across multiple samples. Enlarging the Koopman matrix exacerbates the issue by encouraging dynamic modes to absorb static signals, while

a lower-rank Koopman matrix does not have the capacity to encode the data. To address this, we approximate the Koopman operator *per sample*, assigning each sequence its own Koopman matrix. This design preserves SKD's training procedure while enhancing representational expressiveness, allowing the model to capture fine-grained static information in a single dedicated subspace. Additionally, extraction of disentangled representations is simplified: each coordinate of the static vector is orthogonal, and may cleanly correspond to a distinct semantic factor—or multiple coordinates may jointly describe a single one. A detailed method description, illustrative diagrams, and key differences between SKD and SSM-SKD are provided in App. C.2.

### 3.7 VLM-as-a-tagger/judge

A major barrier to disentanglement evaluation is the reliance on labeled data to verify whether latent dimensions correspond to semantic factors. While feasible in synthetic settings, this becomes impractical in real-world applications, where annotations are costly or unavailable. To overcome this, we introduce a *VLM-based tagger/judge*—a VLM [61] that infers factor values directly from visual inputs. By replacing dataset-specific classifiers with a general-purpose reasoning engine, our approach supports scalable, label-free evaluation across diverse vision modalities. Although current disentanglement methods fail to recover meaningful factors when trained on real-world datasets (see Sec. 4.3), our VLM-based framework enables future progress by removing annotation bottlenecks. We validate its utility in two experiments, described in Sec. 4.4.

**Integration flow.** The VLM is integrated into four benchmark stages: (1) Factor discovery: given an unlabeled dataset, the VLM returns a list of high-level visual factors (e.g., hair color, age); (2) Label assignment: for each factor, the VLM assigns a label to each sample (e.g., "black" or "blonde" for hair color); (3) Latent exploration: in classifier-dependent LES strategies, the VLM serves as a flexible classifier to map latent dimensions to factors; and (4) Evaluation: in metrics requiring classification, the VLM replaces task-specific models, enabling fully automated evaluation. See App. F.2 for a visual summary.

**Implementation details.** Factor discovery is performed via pairwise sample comparisons, prompting the VLM to describe visual differences. These responses are aggregated and clustered by a language model to form canonical factors. Infrequent attributes are discarded, and each factor is labeled as static or dynamic based on inspection of representative sequences. For label space discovery, the VLM is prompted to extract and consolidate observed values into concise, practical label sets. Finally, for judging, the VLM is presented with a sample, a target factor, and a predefined set of possible labels, and is tasked with selecting the most appropriate label, eliminating the need for dataset-specific classifiers. Figures illustrating each functionality are provided in App. F.

## 4 Benchmarking results

In this section, we present a comprehensive quantitative comparison of all methods (Sec. 4.1), a qualitative analysis of selected approaches (Sec. 4.2), and a failure case highlighting the limitations of current multi-factor disentanglement methods (Sec. 4.3). We conclude with an evaluation of our VLM-based method to assess its reliability and accuracy (Sec. 4.4).

### 4.1 Quantitative benchmarking

We evaluate all methods across six datasets using our disentanglement metrics. Assessing multi-factor disentanglement is challenging, as metrics must capture not only successful manipulation of target factors but also the preservation of unrelated ones, leading to measures that can be hard to interpret. To simplify comparison, we aggregate results into a single representative score per dataset

Table 1: Summary of disentanglement metrics across various datasets (higher numbers are better). Bold values denote the best performance for each row.

| Dataset | Sparse-AE | VAE | $\beta$-VAE | MGP-VAE | SKD | SSM-SKD |
|---|---|---|---|---|---|---|
| Sprites | $0.73 \pm 2\text{e-}03$ | $0.58 \pm 3\text{e-}03$ | $0.60 \pm 2\text{e-}03$ | $0.79 \pm 3\text{e-}03$ | $0.75 \pm 1\text{e-}02$ | $\mathbf{0.95 \pm 2\text{e-}03}$ |
| 3D Shapes | $0.85 \pm 9\text{e-}04$ | $0.93 \pm 7\text{e-}04$ | $0.82 \pm 7\text{e-}04$ | $0.65 \pm 2\text{e-}03$ | $0.76 \pm 1\text{e-}03$ | $\mathbf{0.96 \pm 4\text{e-}04}$ |
| dSprites-Static | $0.50 \pm 1\text{e-}03$ | $0.52 \pm 1\text{e-}03$ | $0.70 \pm 2\text{e-}03$ | $0.53 \pm 2\text{e-}03$ | $0.64 \pm 1\text{e-}03$ | $\mathbf{0.83 \pm 2\text{e-}03}$ |
| dSprites-Dynamic | $0.68 \pm 2\text{e-}03$ | $0.63 \pm 1\text{e-}03$ | $0.65 \pm 2\text{e-}03$ | $0.53 \pm 2\text{e-}03$ | $0.64 \pm 4\text{e-}03$ | $\mathbf{0.71 \pm 1\text{e-}03}$ |
| dMelodies-WAV | $0.47 \pm 3\text{e-}03$ | $0.45 \pm 2\text{e-}03$ | $0.52 \pm 2\text{e-}03$ | $0.35 \pm 3\text{e-}03$ | $0.52 \pm 2\text{e-}03$ | $\mathbf{0.55 \pm 2\text{e-}03}$ |
| BMS Air Quality | $0.36 \pm 5\text{e-}03$ | $\mathbf{0.42 \pm 2\text{e-}03}$ | $\mathbf{0.42 \pm 4\text{e-}03}$ | $0.17 \pm 2\text{e-}03$ | $0.36 \pm 9\text{e-}03$ | $\mathbf{0.42 \pm 2\text{e-}03}$ |

Table 2: Leaderboard showing the mean and standard deviation of each model's performance across metrics. Higher values indicate better performance. SKD-based methods lead the board.

| # | M-Swap ↑ | M-GSample ↑ | DCI-M ↑ | DCI-C ↑ | DCI-E ↑ | C-Swap ↑ | C-Sample ↑ | GC-Sample ↑ | 2-Swap ↑ | 2-GSample ↑ |
|---|---|---|---|---|---|---|---|---|---|---|
| 1.0 | SSM-SKD (0.79 ± 0.14) | SSM-SKD (0.81 ± 0.14) | SSM-SKD (0.51 ± 0.37) | SSM-SKD (0.81 ± 0.13) | SSM-SKD (0.80 ± 0.22) | SSM-SKD (0.83 ± 0.14) | SSM-SKD (0.95 ± 0.02) | SSM-SKD (0.96 ± 0.02) | SKD (0.87 ± 0.12) | SKD (0.87 ± 0.11) |
| 2.0 | SKD (0.69 ± 0.06) | SKD (0.70 ± 0.07) | $\beta$-VAE (0.26 ± 0.14) | $\beta$-VAE (0.79 ± 0.04) | SKD (0.58 ± 0.14) | MGP-VAE (0.66 ± 0.26) | $\beta$-VAE (0.95 ± 0.05) | $\beta$-VAE (0.95 ± 0.06) | SSM-SKD (0.86 ± 0.14) | SSM-SKD (0.86 ± 0.13) |
| 3.0 | MGP-VAE (0.68 ± 0.14) | VAE (0.67 ± 0.11) | Sparse-AE (0.25 ± 0.29) | VAE (0.77 ± 0.10) | Sparse-AE (0.55 ± 0.31) | Sparse-AE (0.56 ± 0.27) | SKD (0.94 ± 0.04) | SKD (0.95 ± 0.03) | MGP-VAE (0.79 ± 0.09) | $\beta$-VAE (0.74 ± 0.14) |
| 4.0 | VAE (0.67 ± 0.11) | $\beta$-VAE (0.67 ± 0.04) | VAE (0.24 ± 0.33) | Sparse-AE (0.77 ± 0.09) | $\beta$-VAE (0.53 ± 0.24) | SKD (0.53 ± 0.07) | Sparse-AE (0.91 ± 0.11) | Sparse-AE (0.93 ± 0.08) | $\beta$-VAE (0.73 ± 0.14) | VAE (0.70 ± 0.15) |
| 5.0 | Sparse-AE (0.67 ± 0.07) | Sparse-AE (0.64 ± 0.04) | MGP-VAE (0.17 ± 0.22) | SKD (0.63 ± 0.06) | VAE (0.45 ± 0.29) | $\beta$-VAE (0.49 ± 0.22) | VAE (0.90 ± 0.06) | VAE (0.93 ± 0.06) | VAE (0.70 ± 0.15) | MGP-VAE (0.69 ± 0.09) |
| 6.0 | $\beta$-VAE (0.66 ± 0.03) | MGP-VAE (0.57 ± 0.06) | SKD (0.14 ± 0.10) | MGP-VAE (0.47 ± 0.14) | MGP-VAE (0.42 ± 0.27) | VAE (0.43 ± 0.35) | MGP-VAE (0.77 ± 0.04) | MGP-VAE (0.74 ± 0.04) | Sparse-AE (0.70 ± 0.12) | Sparse-AE (0.66 ± 0.06) |

$S_{m,d} = \frac{1}{K_d} \sum_{k=1}^{K_d} s_{m,d,k}$, where $S_{m,d} \in [0,1]$ is the aggregated score for model $m$ on dataset $d$, $s_{m,d,k}$ is the normalized value of metric $k$, and $K_d$ is the number of applicable metrics. This ensures comparability across datasets while weighting all metrics equally. Reported $\pm$ values denote standard errors over five evaluation runs. Details appear in App. D, with full results in App. G. Summary scores are shown in Tab. 1.

To capture metric-specific trends, we also report a *per metric leaderboard* summarizing each model's performance averaged over datasets where a metric applies: $L_{m,k} = \frac{1}{|\mathcal{D}_k|} \sum_{d \in \mathcal{D}_k} s_{m,d,k}$, where $L_{m,k}$ is the leaderboard score of model $m$ on metric $k$ and $\mathcal{D}_k$ is the set of valid datasets. Each column represents a metric and each row a rank level, as shown in Tab. 2, highlighting which models perform best under specific evaluation criteria and complementing the aggregated view in Tab. 1.

**Model comparison.** The final results reveal several key insights. First, our method, SSM-SKD, consistently achieves state-of-the-art performance across a diverse set of datasets. It ranks first on 4 out of 6 benchmarks. For example, on the Sprites dataset, it outperforms the current strong baseline with an approximate 16% improvement. Moreover, on the more challenging dSprites synthetic datasets, it achieves a significant performance gap over prior methods, demonstrating both robustness and effectiveness. Second, while our implementations of baseline methods such as Sparse-AE and $\beta$-VAE occasionally achieve competitive results, they still lack consistency across datasets. However, they are simple, and we suggest that incorporating recent advances in sparse modeling and improved disentanglement techniques for VAEs may hold promise for future research.

**Dataset analysis.** While certain datasets, such as Sprites and 3D Shapes, are nearing saturation, others remain far from solved. For example, no method exceeds a score of 0.85 on either variant of the dSprites dataset, indicating significant room for improvement even on synthetical setups. Similarly, performance on dMelodies-WAV and BMS Air Quality remains suboptimal across all evaluated models, suggesting further opportunities for methodological advancement. Overall, we hope our comprehensive analysis provides a clear snapshot of current progress and will encourage future efforts to close the remaining performance gaps.

## 4.2 Qualitative benchmarking

To qualitatively assess disentanglement, we visualize the effect of swapping individual factors between two reference samples. In Fig. 2, we present an example using two reference samples (Sample 1/2). For each method (SSM-SKD and MGP-VAE), we sequentially transfer a single semantic factor from Sample 2 to Sample 1. The considered factors include skin tone, pants color, shirt color, hair color, and movement dynamics. This visualization offers intuitive insight into each model's ability to isolate and manipulate specific factors. For example, MGP-VAE fails to correctly swap hair color, erroneously changing it to green instead of purple. While qualitative results are useful for identifying model failure cases, they may also be misleading—particularly for datasets with many attributes or when models exhibit stochastic behavior, making cherry-picking or incidental inconsistencies more likely. Therefore, we recommend treating qualitative evaluation as a complementary diagnostic tool to support and contextualize quantitative results, rather than as a primary evaluation metric.

## 4.3 Failure case: real-world data

While our benchmark demonstrates that existing methods can operate on realistic data modalities such as time series and audio, both the quantitative results and the following experiment indicate that

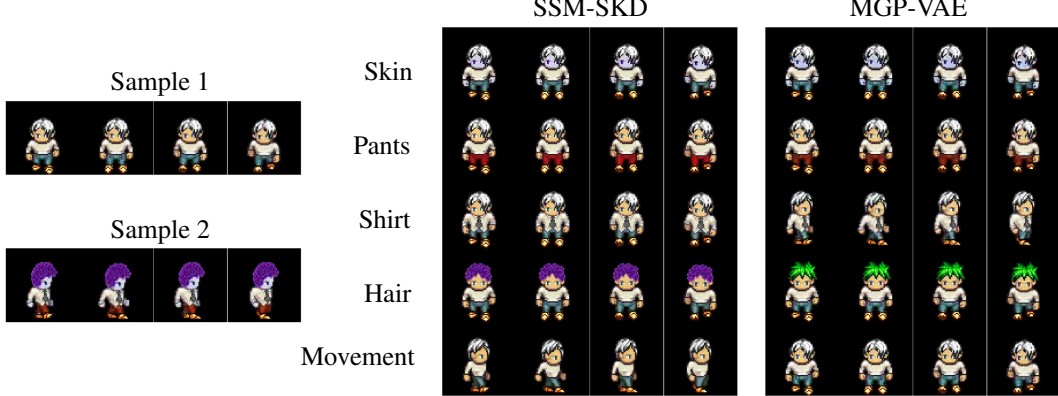

Figure 2: Qualitative factor-swapping between two reference samples (left). For each row, a single factor from Sample 2 is swapped into Sample 1. Results are shown for two models: SSM-SKD (middle block) and MGP-VAE (right block). Factors include skin tone, pants color, shirt color, hair color, and movement dynamics (top to bottom).

their performance remains suboptimal, with considerable room for improvement. To further analyze this issue, we evaluate the current state-of-the-art model on real-world video data. Specifically, we train SSM-SKD on the VoxCelebOne dataset [42], which contains in-the-wild talking face videos. For qualitative analysis, Fig. 3 presents two original samples (row 1), their reconstructions (row 2), and a sex-swapped variant (row 3). The results reveal that the current model struggles to capture high-level semantic attributes. Although a correct swap can be visually perceived to some extent, the output remains blurry, and other factors, such as background, remain entangled. While these results are shown for SSM-SKD, similar limitations are observed across all existing methods, which are largely based on VAE architectures [28]. A primary reason for this failure is the reliance on similar encoder-decoder designs based on AE/VAE backbones, which are known to suffer from disentanglement-reconstruction trade-offs [10]. We believe these findings help clarify the current limitations of multi-factor disentanglement models and will hopefully encourage future efforts to close the remaining performance gaps. We hypothesize that utilizing paradigms such as diffusion models [16], GANs [49], and hierarchical VAE architectures [56] could be fruitful for improving both generation quality and disentanglement.

### 4.4 Assessing a VLM-as-a-tagger/judge

In this section, we validate the applicability of the VLM-based tagger/judge both in the sense of applicability to real-world problems and alignment to ground truth judgment.

**Performance on a real-world dataset.** Our goal is to enable benchmarking of disentanglement methods on real-world datasets. As shown in Sec. 4.3, current methods often fall short in this setting, however, practical evaluation tools can be helpful for future methical development. We apply our VLM-based tagger to the VoxCelebOne dataset. Example frames are shown in Fig. 4. Tab. 3 presents selected semantic factors and their identified values, collected fully autonomously by the VLM. The VLM effectively captures key features and labels across data variations. We validate its zero-shot tagging accuracy against human annotations on 100 samples. Results show near-perfect performance

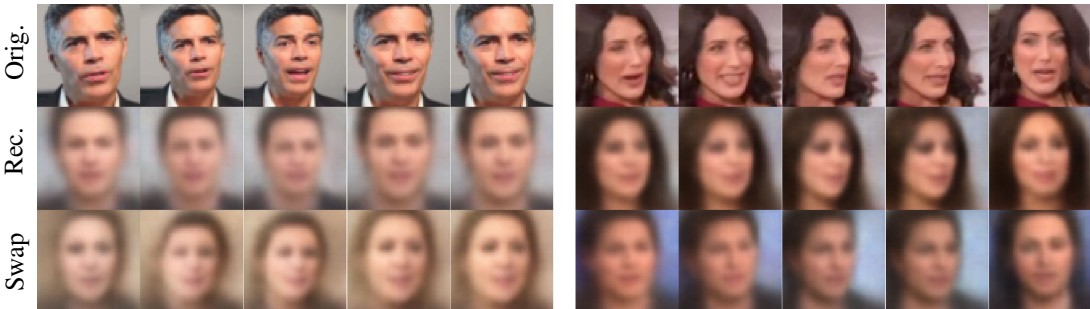

Figure 3: SSM-SKD results on VoxCelebOne: Row 1 shows the original samples (Orig.), row 2 the reconstructions (Rec.), and row 3 an attribute swap (Swap).

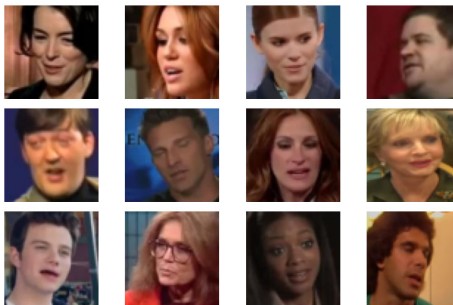

Figure 4: Sampled frames from VoxCelebOne.

Table 3: Semantic factors with their possible values and VLM annotation accuracy.

| Factor | Values | Accuracy |
|---|---|---|
| Background | red, green, ... | 0.92 |
| Hair style | short, wavy, ... | 0.81 |
| Hair color | black, brown, ... | 0.84 |
| Sex | male, female | 1.00 |
| Lighting | bright, dim, ... | 0.72 |
| Shirt color | green, blue, ... | 0.63 |

for sex and background, with strong accuracy for hair-related factors. Lower scores for lighting and shirt color are likely due to visual ambiguity, which also affects human judgment. Although the labeling process is not perfect, it enables a capability that was previously unavailable: flexible supervised evaluation on real-world datasets. Further, it is worth noting that users can introduce factors of variation that the model may have overlooked; our pipeline is designed to retrieve their corresponding values. Full results can be found in App. F.

**Alignment with full setup evaluation.** To validate the reliability of the fully end-to-end automatic evaluation, we assess whether the final method rankings align with those obtained under a setup with full access to ground truth factors and labels. To simulate such an environment, we utilize the final evaluation on the Sprites and 3D Shapes datasets and consider the average of four metrics: M-Swap, DCI-C, DCI-E, and DCI-M. We compare the rankings produced by the fully automatic setup (using VLM) with those derived from ground truth supervision. We present both the graphical correlation plots and the corresponding Spearman correlation values in Fig. 5. The results demonstrate an almost perfect alignment between the automatic VLM-based evaluation and the fully supervised ground truth evaluation. This highlights the validity and robustness of the VLM evaluation flow as a strong proxy for fully labeled assessments.

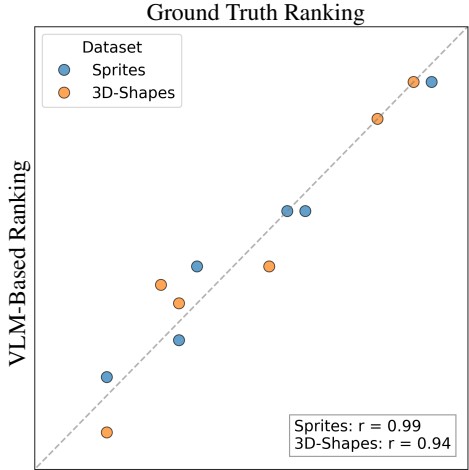

Figure 5: Agreement between VLM-based and supervised evaluations.

## 5   Discussion

This work takes a significant step toward standardized and scalable benchmarking of multi-factor sequential disentanglement. Our benchmark introduces diverse realistic and synthetic datasets, novel evaluation protocols, and implementations of leading methods. Our proposed SSM-SKD model provides a strong baseline, but clear gaps remain in all current models, especially in capturing complex, high-level semantics. We further demonstrate the importance of the LES module, and highlight the value of VLMs in enabling tagging and evaluation on unlabeled real-world datasets.

An interesting trend in our results (Tab. 1) is that the vanilla VAE occasionally achieves the highest score. This indicates limitations of current methods rather than benchmark bias—when all models struggle, absolute scores are low, though in some datasets (e.g., 3D Shapes) the VAE meaningfully disentangles factors. More broadly, this reflects the field's focus on visual domains, with audio and time series models still underdeveloped. To address this, we introduce dMelodies-WAV and BMS Air Quality as challenging benchmarks from underrepresented modalities, promoting progress beyond vision. Our benchmark thus highlights both the strengths and gaps of current methods, emphasizes the need for modality-agnostic approaches, and establishes a foundation for models that generalize across diverse domains. Further discussion appears in App. E.

## Acknowledgments

This research was partially supported by the Lynn and William Frankel Center of the Computer Science Department, Ben-Gurion University of the Negev, an ISF grant 668/21, an ISF equipment grant, and by the Israeli Council for Higher Education (CHE) via the Data Science Research Center, Ben-Gurion University of the Negev, Israel.

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

# A   Codebase

We provide a modular, extensible, and fully configurable codebase designed to support large-scale benchmarking of disentanglement methods in sequential domains. The codebase is organized into three main modules: *Datasets*, *Methods*, and *Evaluation*, each designed to ensure clarity, reproducibility, and ease of integration with new components. An overview of the codebase structure and execution flow is illustrated in Fig. 1.

**Datasets.**   Dataset construction is handled via dedicated scripts for each modality (e.g., video, audio, time series), which store outputs in a standardized `HDF5` format. Datasets such as *dSprites* and *3D Shapes* are originally static, providing the full combinatorial state space of factor configurations. To extend these into the sequential domain, we include procedural generators (App. B.1) that synthesize dynamic sequences by controlling independent factors over time using composable dynamic modifiers. Each sample is labeled with a factor-annotated metadata dictionary, making it compatible with all evaluation modules.

In addition, we provide an automatic annotation component based on VLMs, which enables factor supervision without relying on ground truth labels. This component supports both zero-shot and few-shot configurations, allowing users to either supply a small number of examples or rely purely on textual prompts. Beyond label prediction, the VLM module is capable of discovering the factor space itself (i.e., identifying which attributes vary across samples), as well as the label space (i.e., enumerating the possible values for each factor). This enables flexible and scalable annotation pipelines across modalities, especially in domains where manual labeling is expensive or ambiguous.

**Methods.**   All models are implemented as subclasses of a shared `AbstractModel` class, which enforces a standardized interface to ensure compatibility with evaluation modules and visualizers. Each model must implement the following core methods: `encode`, which maps input data to a latent representation; `decode`, which reconstructs inputs from latent codes; `latent_vector`, which extracts a flat latent representation for downstream analysis; `latent_dim`, which returns the dimensionality of the latent space; and `forward`, which defines the full end-to-end computation from input to output, including any auxiliary predictions or regularization terms. Model behavior and training configurations are fully specified via declarative YAML files using OmegaConf, enabling users to customize architectures, losses, and schedules without modifying source code.

The training procedure is controlled by the `Trainer` module, which standardizes optimization, logging, checkpointing, and learning rate scheduling. Each trainer subclass corresponds to a specific model (or model family) and encapsulates its unique loss formulation, training logic, and regularization schemes. During training, input batches are processed through the model's `forward` method, and losses are computed with respect to both reconstruction quality and disentanglement objectives (e.g., mutual information penalties, eigenvalue regularization, or predictive consistency). The trainer logs metrics and intermediate outputs at configurable intervals and supports automatic saving of checkpoints, early stopping, and resuming from previous runs.

**Evaluation.**   The core of our benchmark lies in the `EvaluationManager`, which orchestrates the evaluation pipeline. Each evaluation run is composed of multiple `Evaluator` modules, which may implement different metrics or analysis procedures. Evaluators can access model latents, reconstructed outputs, classifier predictions, or synthesized samples to compute their metrics.

Central to the evaluation design are three key components: the `LatentExplorer`, the `Judge`, and the `Predictor`. The `LatentExplorer` discovers mappings between latent dimensions and semantic factors, using either supervised classifiers, swap-based interventions, or custom exploration strategies. The `Judge` module evaluates the alignment between generated outputs and target attributes. It supports both classifier-based judges (i.e., pretrained neural classifiers) and VLM-based judges that enable zero- and few-shot evaluation without requiring ground truth annotations. The `Predictor` module, typically implemented as a Gradient Boosting Classifier, is used in modularity-based metrics to measure how well latent representations can predict each factor independently.

Our benchmark includes a broad set of evaluation metrics. These include intervention-based metrics such as *MultiFactor* and *TwoFactor* swap/sample evaluations, which assess how manipulating latent subsets affects factor-specific predictions. We also include `Consistency` metrics, which evaluate whether static and dynamic factors are preserved or appropriately altered across time. Finally, the

`DCI` metrics measure modularity (disentanglement of factor influence), compactness (concentration of factor information), and explicitness (predictability of factors from latent space). All results are logged as structured tables and visualizations, facilitating systematic and interpretable benchmarking.

# B Additional data details

## B.1 Static to dynamic generators

In datasets like dSprites and 3D Shapes, dynamic video sequences are synthetically generated by extending the original static datasets with controlled, factor-wise temporal transformations.

The process follows a modular design, consisting of three main components:

1. Each dynamic factor (e.g., scale, orientation) originates from a static factor that was present in the original dataset. Dynamic factors are associated with multiple pre-defined sequences that describe different ways in which the factor's value can evolve over time across frames. For example, in 3D Shapes, the scale factor may oscillate smoothly between small and large values, while orientation might rotate cyclically.

2. We then define the full state-space of possible sequences. Each sequence is specified by:
   - A static configuration - fixed values for the entire sequence)
   - A dynamic configuration - values that changes over time)

3. Finally, we render the actual video sequences for each sample within the state space. For each sampled (static & dynamic) state, we generate a full sequence of frames by repeatedly mapping factor values into rendered images.

This architecture is designed with several key principles in mind. Controlled dynamics ensure that only specific factors are allowed to vary during each sequence, enabling clear attribution of observed motions to known underlying causes. The framework supports flexible generator assignment, where each dynamic factor can evolve according to different temporal patterns, allowing a rich diversity of dynamic sequences to be produced. To support evaluation and supervision, label alignment guarantees that every sequence is annotated systematically, providing detailed information about both static and dynamic factors across time. Finally, by combining all static configurations with all possible dynamic sequences, the process results in a rich state space, where even a relatively small number of original factors can generate a vast and varied dataset of temporally coherent video sequences.

## B.2 Datasets

- **BMS Air Quality (time series)** [13]: A real-world dataset of 24-step sequences with 13 features, annotated with 5 static environmental and temporal factors: station, year, month, day, and season. We collect and annotate this dataset to support sequential disentanglement tasks.

- **dMelodies-WAV (audio)** [48, 58]: A synthetically generated dataset of audio waveforms labeled with one static factor (instrument) and five dynamic musical attributes (tonic, scale, rhythm bar1, arpeggio chord1, arpeggio chord2). Despite its synthetic nature, the audio is perceptually realistic, and we provide both the dataset and its generating labels. We utilized the dMelodies [48] symbolic music disentanglement dataset and created an equivalent real-world raw waveform music dataset using MIDI-DDSP [58].

- **dSprites-Static (video)** [24]: Consists of 16-frame sequences of synthetic shapes with static attributes (color, shape, position) and dynamic transformations (scale speed, rotation speed). We built a generic video generator on top of the original dSprites image generator. We contribute this tool to enable users to create new, dynamic variations.

- **dSprites-Dynamic (video)** [24]: Contains 12-frame sequences with static visual attributes (color, shape, orientation, scale) and dynamic positional movement along the X and Y axes. This dataset also uses our video generator infrastructure.

- **3D Shapes (video)** [27]: A dataset of 10-frame rendered sequences featuring static factors (floor hue, wall hue, object hue, shape) and dynamic attributes (scale and orientation). Similar to the dSprites variants, we extend the original image generator with temporal dynamics and make the video generator publicly available.

- **Sprites (video)** [34]: Composed of 8-frame animated character sequences labeled with one dynamic factor (movement) and four static appearance factors (body, bottom, top, hair). We use an existing variant of this dataset for consistency and ease of comparison.

- **VoxCelebOne (video)** [42] derived from the VoxCelebOne audio-visual corpus. Here we focus solely on the visual modality. Each sample consists of a short video segment depicting a speaking individual, extracted from real-world interview recordings on YouTube. The dataset captures a wide diversity of speakers across ethnicities, accents, and age groups, and includes variation in pose, expression, lighting, and background conditions. For our purposes, we preprocess each face track into fixed-length sequences of cropped face images. Sequences are standardized to facilitate disentanglement analysis under real-world variability.

Table 4: Datasets supported in our benchmark. Each entry includes dataset type, split sizes, sequence characteristics, and disentangled factors.

| Dataset | Type | Train / Val / Test | Seq. Len. | Features | Factors (Type, #Classes) |
|---|---|---|---|---|---|
| BMS Air Quality [13] | Time-series | se-12272 / 2630 / 2630 | 24 | 13 | Station (static, 12), Year (static, 5), Month (static, 12), Day (static, 31), Season (static, 4) |
| dMelodies-WAV [48, 58] | Audio | 11289 / 2419 / 2420 | 48000 | – | Instrument (static, 4), Tonic (dynamic, 12), Scale (dynamic, 3), Rhythm Bar1 (dynamic, 28), Arp Chord1 (dynamic, 2), Arp Chord2 (dynamic, 2) |
| dSprites-Static [24] | Video | 21772 / 4666 / 4666 | 16 | (3, 64, 64) | Color (static, 9), Shape (static, 3), PosX (static, 8), PosY (static, 8), Scale Speed (dynamic, 6), Rotation Speed (dynamic, 3) |
| dSprites-Dynamic [24] | Video | 20412 / 4374 / 4374 | 12 | (3, 64, 64) | Color (static, 9), Shape (static, 3), Scale (static, 6), Orientation (static, 5), PosX Dynamic (dynamic, 6), PosY Dynamic (dynamic, 6) |
| 3D Shapes [27] | Video | 50400 / 10800 / 10800 | 10 | (3, 64, 64) | Floor Hue (static, 10), Wall Hue (static, 10), Object Hue (static, 10), Shape (static, 4), Scale Dynamic (dynamic, 6), Orientation Dynamic (dynamic, 3) |
| Sprites [34] | Video | 8164 / 1750 / 1750 | 8 | (3, 64, 64) | Movement (dynamic, 9), Body (static, 6), Bottom (static, 6), Top (static, 6), Hair (static, 6) |
| VoxCelebOne [42] | Video | 153386 / 500 | 10 | (3, 64, 64) | Background Color (static, 16), Hair Style (static, 11), Hair Color (static, 8), Sex (static, 2), Lighting (dynamic, 7), Shirt Color (dynamic, 10), Age Group (static, 5), Skin Color (static, 3), Glasses (static, 2), Facial Hair (static, 2), Earrings (static, 2) |

### B.2.1 dMelodies-WAV

The dMelodies-WAV dataset extends the dMelodies [48] symbolic music disentanglement dataset into the raw audio domain, enabling novel research directions in disentangled representation learning for music. The original dMelodies dataset consists of 2-bar monophonic melodies generated from symbolic notation, governed by a well-defined set of independent factors including musical scale, chord progression, rhythm, and arpeggiation direction. Each melody adheres to a structured I–IV–V–I chord progression and is constructed using discrete rhythmic and harmonic patterns, ensuring control and interpretability. To bring this symbolic dataset into the waveform domain, we synthesized a subset of dMelodies using the MIDI-DDSP [58] neural audio synthesis model, producing realistic-sounding audio sequences across four different instruments (violin, trumpet, saxophone, and flute). The resulting dMelodies-WAV dataset includes both global factors (instrument, tonic, and scale) and local factors (rhythm and arpeggiation direction per chord), providing a rich structure for studying disentanglement in raw waveform music. By bridging symbolic and raw music representations,

dMelodies-WAV introduces the first multi-factor disentanglement benchmark in raw waveform music, opening new avenues for evaluating multi-scale and hierarchical multi-factor sequential disentanglement methods.

### B.2.2 BMS Air Quality

The BMS Air Quality dataset is adapted for disentanglement analysis by structuring it into fixed-length temporal sequences. Raw hourly records, collected between 2013 and 2017 from multiple monitoring stations across Beijing, are preprocessed by grouping measurements according to station, year, month, and day, resulting in sequences of length 24 corresponding to full daily records.

Environmental features, including pollutant concentrations (e.g., $PM2.5$, $PM10$, $SO_2$, $NO_2$, $CO$, $O_3$), meteorological variables (e.g., temperature, pressure, dew point, rainfall, wind speed), and wind direction (encoded via sine and cosine transformations), are normalized across the dataset. Missing entries are interpolated linearly to ensure continuity within sequences.

Each daily sequence is further labeled with attributes such as the recording station, year, month, day, and climatological season, where seasons are assigned based on date. The resulting dataset provides structured samples and is partitioned into training, validation, and test subsets, with complete metadata annotations for all categorical factors.

# C Additional methods details

We summarize bellow the different methods:

- **Sequential VAE [28]**: An extension of the AE with variational inference, introducing generative capabilities absent in the standard AE.

- **Sequential $\beta$-VAE [23]**: Builds on the VAE by introducing a $\beta$-coefficient to enforce stronger factorization in the latent space, promoting disentanglement.

- **Sequential Sparse-AE [47]**: Extends the AE with a larger latent space and sparsity constraints, encouraging interpretability and factor disentanglement.

- **MGP-VAE [8]**: A structured VAE that uses Gaussian processes with fractional Brownian motion and Brownian bridges to disentangle static and dynamic features over time.

- **SKD [6]**: A Koopman operator-based model that employs spectral loss functions to disentangle multiple latent factors in sequential data.

- **SSM-SKD (Sec. 3.6)**: Our improved variant of SKD, which incorporates a single static mode constraint and enhanced latent space extraction to achieve more efficient disentanglement.

Table 5: Availability and reproducibility of code before and after our benchmarking. "Partially" indicates that results could not be reproduced in the original environment without significant adaptation.

| Method | Code Available Before Benchmark | Reproducible in Original Environment | Reproducible in Our Benchmark |
|---|---|---|---|
| VAE | No | - | Yes |
| $\beta$-VAE | No | - | Yes |
| Sparse-AE | No | - | Yes |
| MGP-VAE | Yes | Partially | Yes |
| SKD | Yes | Partially | Yes |
| SSM-SKD | - | - | Yes |
| DDPAE | Yes | No | No |
| FAVAE | No | No | No |

## C.1 Hyperparameter grid search

We conducted a grid search to identify stable and comparable configurations across methods (Tab. 6). For each method, we defined a compact discrete set of key hyperparameters—such as latent dimensionality, hidden layer sizes, and regularization weights—chosen to balance representational capacity and computational efficiency. From this space, we randomly sampled approximately 100 combinations of parameter values.

Model selection was performed independently for each dataset and method by selecting the best-performing configuration based on predictor-based LES and evaluation over the validation splits. Each method's configuration files define the best-performing hyperparameter values found in our grid search along with default values for any additional hyperparameters.

## C.2 Our method: SSM-SKD

### C.2.1 Motivation

By design SKD constrains all static modes to have eigenvalues which are close to 1. However, we may also consider having just a single static mode with an eigenvalue which is close to 1. This method, which we call **Single Static Mode SKD**, may possibly aid in introducing further constraints on the representation of static factors. Furthermore, it forces orthogonality on the encoding of static factors in the Koopman matrix, since all static factors are encoded in a single eigenvector of orthogonal coordinates.

Table 6: Grid search parameter value sets for each method

| Method | Parameter | Value Set |
|---|---|---|
| Sparse-AE | `latent_dim` | {64, 128, 256, 512} |
| | `sparsity_weight` | {0.01, 0.1, 1.0} |
| | `hidden_dims` | {(32, 64, 128, 256), (64, 128, 256, 512), (128, 256, 512, 1024)} |
| VAE | `latent_dim` | {64, 128, 256, 512} |
| | `hidden_dims` | {(32, 64, 128, 256), (64, 128, 256, 512), (128, 256, 512, 1024)} |
| $\beta$-VAE | `latent_dim` | {64, 128, 256, 512} |
| | `beta` | {2, 3, 5, 8} |
| | `hidden_dims` | {(32, 64, 128, 256), (64, 128, 256, 512), (128, 256, 512, 1024)} |
| MGP-VAE | `NUM_FEA` | {4, 5, 6} |
| | `FEA_DIM` | {2, 3} |
| | `FEA` | {bb, bb2} |
| | `fac` | {0.1, 0.5, 0.9} |
| | `kl_beta` | {2.0, 3.0} |
| SKD | `k_dim` | {26, 40, 54} |
| | `hidden_dim` | {80, 90, 110, 140, 180} |
| | `w_rec` | {11.0, 12.0, 13.0, 14.0, 15.0, 16.0} |
| | `w_pred` | {0.25, 1.0, 4.0} |
| | `w_eigs` | {0.25, 1.0, 4.0} |
| | `static_size` | {6, 7, 8, 9, 10, 11} |
| | `static_mode` | {ball, norm} |
| | `dynamic_thresh` | {0.125, 0.25, 0.275, 0.425, 0.5, 0.575, 0.725, 0.75, 0.875} |
| SSM-SKD | `k_dim` | {15, 16, 19, 20, 23, 24, 27, 28, 31, 32, 43, 54} |
| | `hidden_dim` | {80, 90, 120, 130, 160, 170, 200, 210} |
| | `w_pred` | {0.25, 1.0, 4.0} |
| | `w_eigs` | {0.25, 1.0, 4.0} |
| | `dynamic_thresh` | {0.125, 0.25, 0.275, 0.425, 0.5, 0.575, 0.725, 0.75, 0.875} |

### C.2.2 Architecture in relation to SKD

We begin by defining a single static eigenvalue (static size 1) and attempt to reproduce the results of SKD using this setting. However, we encounter the shortcut problem [54]: when the Koopman matrix size $K$ is small ($K \leq 8$), the model fails to reconstruct. On the other hand, when $K$ is large, the model is unable to achieve the desired static-dynamic disentanglement, as it exploits the higher capacity of the additional modes to encode static information.

To address this, we modify the Koopman operator approximation from a per batch to a per instance formulation. Specifically, we solve a least-squares problem for each instance in the batch: for every $0 \leq i \leq N - 1$, we compute a solution to the linear system

$$Z_{i,0:T-2}\mathcal{K}_i = Z_{i,1:T-1},$$

where $Z$ denotes the latent representation from the encoder, $i$ indexes instances in the batch, and $T$ is the sequence length. The colon operator denotes slicing along the temporal dimension. As a result, we obtain $N$ Koopman matrices $\{\mathcal{K}_i\}$, one for each instance in the batch.

In SSM-SKD we redefine $\mathcal{K}$ as a tensor that concatenates all per instance matrices $\{\mathcal{K}_i\}$ along a new batch dimension (dimension index 0).

### C.2.3 Latent space extraction in relation to SKD

SKD extracts a latent space representation at the batch level. Consistent with the principles of dynamic mode decomposition (DMD) [50], SKD defines the Koopman latent representation of an instance along particular modes as multiplication of its latent matrix $Z_i$ - where $i$ indexes the instance within the batch - by a submatrix of the eigenvector matrix of $\mathcal{K}$ which corresponds to the subspace of selected modes. Therefore, to perform attribute swapping between two instances in the batch, each instance's latent matrix is first projected into the Koopman eigenbasis via multiplication with the eigenvector matrix. Then, the components corresponding to the desired modes are swapped. Finally, each swapped representation is projected back into the original latent space by multiplying with the inverse of the eigenvector matrix, and subsequently passed through the decoder to produce the output.

In contrast, SSM-SKD approximates the Koopman operator individually per instance, making SKD's batch-level latent extraction incompatible. Instead, for each instance, we isolate its static latent representation by projecting its latent matrix onto the subspace spanned by the static mode - achieved via multiplication with the corresponding submatrix of the eigenvector matrix, followed by multiplication with the corresponding submatrix of the inverse of the eigenvector matrix. The dynamic latent representation is similarly obtained by projecting onto the subspace spanned by the dynamic modes. We treat the $K$ feature coordinates from the static projections and the $K$ feature coordinates from the dynamic projections as the static and dynamic channels of the instance, respectively.

To perform attribute swapping under this formulation, we simply exchange the contents at the corresponding static and/or dynamic channels between two instances. The final representations - obtained by summing the modified static and dynamic latent representations - are then passed through the decoder to produce the output.

We provide a visual comparison between our method and SKD [6] in Fig. 6. Let $Z$ denote the latent tensor of the batch. This tensor has shape $(N, T, K)$, where $N$ is the batch size, $T$ is the temporal length of the sequences, and $K$ (the size of Koopman matrices) is the dimensionality of the latent features at each time step.

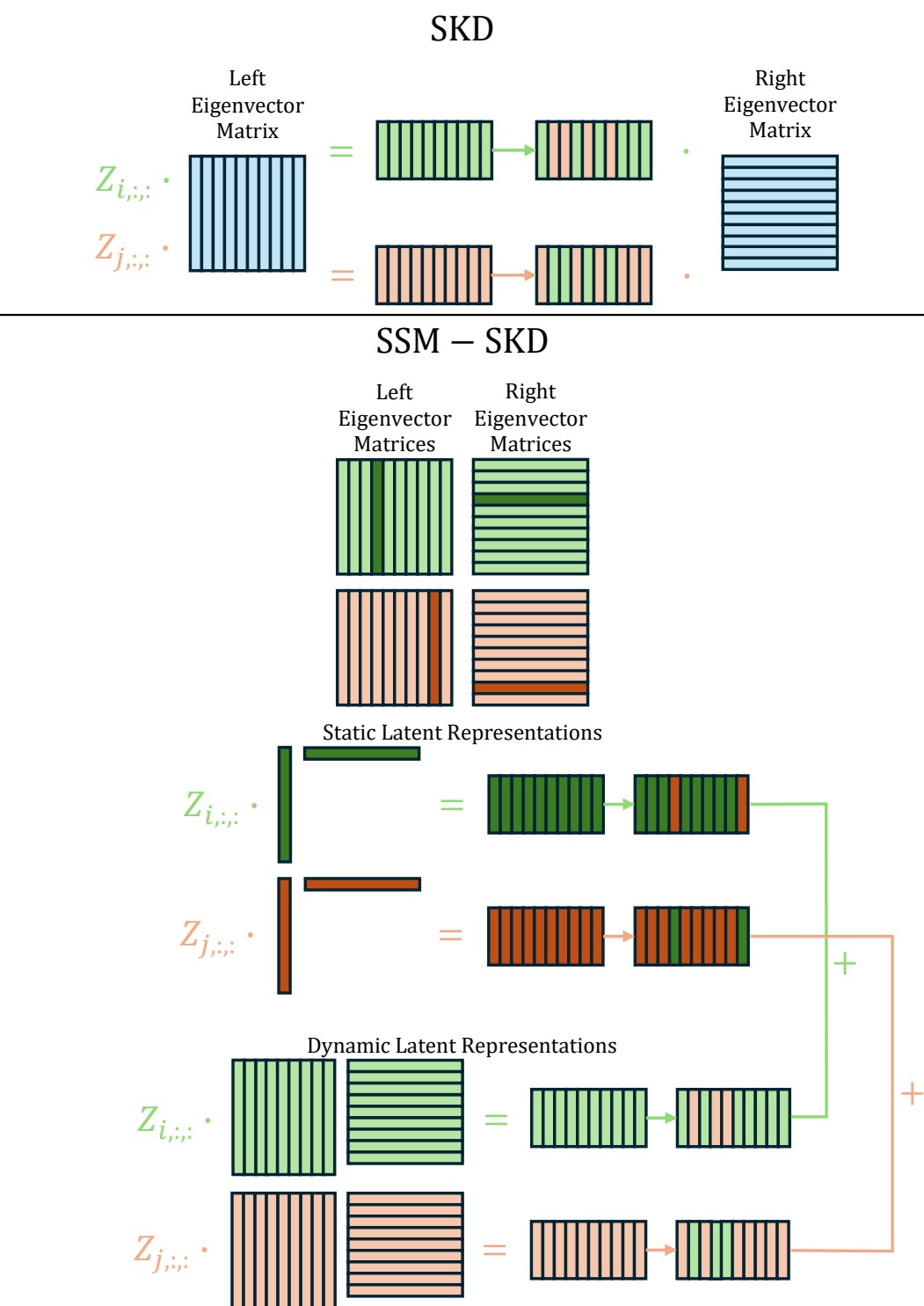

Figure 6: Latent space extraction and swap in SKD and in SSM-SKD

# D   Additional metrics details

Below we will detail each metric in our benchmark. We encourage reareds to seek further information regarding disentanglament metrics and their taxonomy in the great survey of [11]. We define in Sec. 3.1 the problem formulation and notations that will be used along this section.

## D.1   Two-factor Swap (2-Swap)

Two-factor swapping is a long-standing benchmark for evaluating two-factor sequential disentanglement methods [2]. In this task, the data is assumed to be governed by two types of factors: static (e.g., identity) and dynamic (e.g., motion). Given two samples $x_1$ and $x_2$ drawn from a dataset with corresponding, the objective is to disentangle their representations and swap one factor while keeping the other fixed. Formally, let $z_1 = (s_1, d_1)$ and $z_2 = (s_2, d_2)$ denote the latent representations of $x_1$ and $x_2$, where $s$ and $d$ correspond to the static and dynamic components, respectively. The decoder is then used to reconstruct cross-composed samples: $\hat{x}_1 = \text{dec}(s_1, d_2)$ and $\hat{x}_2 = \text{dec}(s_2, d_1)$, where $\text{dec}$ is the decoding module of the disentanglement model.

A successful swap is one in which the reconstructed samples accurately preserve the intended static factor from one sample and the dynamic factor from the other. For example, $\hat{x}_1$ should retain the identity (static component) of $x_1$ while exhibiting the motion (dynamic component) of $x_2$. We assume access to ground truth labels for both static and dynamic attributes. To evaluate the swap, we measure the accuracy of the preserved factors using a pre-trained classifier or a VLM module. Specifically, we compute the accuracy of the static factor as $\hat{y}_1^{\text{static}} = y_1^{\text{static}}$, where $\hat{y}_1^{\text{static}}$ is predicted from $\hat{x}_1$. Similarly, we evaluate the dynamic factor as $\hat{y}_1^{\text{dynamic}} = y_2^{\text{dynamic}}$, reflecting that $\hat{x}_1$ should exhibit the dynamics of $x_2$. The same evaluation is applied symmetrically to $\hat{x}_2$. This task provides both a qualitative and quantitative assessment of whether the model has successfully learned to disentangle the latent space into interpretable and independent two-factors.

## D.2   Two-factor Generation and Swap (2-GSample)

This metric is similar in spirit to the 2-Swap evaluation but differs in a key aspect. Instead of swapping static and dynamic components between two distinct samples, we operate on a single input sample. Specifically, we extract its disentangled latent representation and then replace either the static or dynamic component with a newly sampled one from the prior or a predefined distribution. This setup allows for evaluating both the generative quality and the disentanglement capacity of the model, offering a broader and more flexible assessment. A successful generative swap should preserve the unaltered factor (either static or dynamic) while meaningfully changing the other. Formally, let $x_1$ be a sample with corresponding static and dynamic labels $y_1 = (y_1^{\text{static}}, y_1^{\text{dynamic}})$. We encode $x_1$ to obtain its disentangled representation $z_1 = (s_1, d_1)$. Then, we sample a new component $\hat{s}_1$ or $\hat{d}_1$ to construct a modified latent representation $\hat{z}_1 = (\hat{s}_1, d_1)$ or $(s_1, \hat{d}_1)$, respectively. The new sample is generated via decoding: $\hat{x}_1 = \text{dec}(\hat{z}_1)$. Evaluation is performed by measuring the accuracy of the swapped and preserved factors using classifiers or VLMs, following the same protocol as in the 2-Swap evaluation.

## D.3   Multi-factor Swap (M-Swap) and Multi-factor Generation and Swap (M-GSample)

This metric extends the two-factor evaluation to a more general multi-factor setting. We assume a set of semantic factors $F$, composed of two disjoint subsets of static and dynamic factors, denoted by $F = S_F \cup S_D$, with access to corresponding ground truth labels. For each individual factor $f_i \in F$, we perform a procedure analogous to the two-factor swapping setup. In the swap protocol, we freeze the latent representation corresponding to a selected factor $f_i$ and randomly modify the representations of all other factors. We then evaluate whether (1) the frozen factor remains unchanged, and (2) the other factors have been successfully altered. An ideal disentanglement model should exhibit near-random classification accuracy for the swapped factors and $100\%$ accuracy for the frozen factor. This process is repeated independently for each $f_i \in F$. An example of this evaluation protocol is illustrated in the supplementary material files. A similar procedure is applied in the generative swap setting, where new values are sampled for all factors except the frozen one.

Since the resulting evaluation table is high-dimensional and difficult to interpret at a glance, we propose a distilled single-score summary. First, we compute the average accuracy of the diagonal entries, which reflect the preservation of the frozen factor and should ideally be $100\%$. Next, we evaluate the effectiveness of changing the non-frozen factors by computing the average deviation of the off-diagonal entries from their respective noise floors. The noise floor for each factor is precomputed based on the number of classes it spans. Finally, we average the diagonal accuracy and the normalized off-diagonal deviation to obtain a single overall score. A precise implementation of this scoring protocol is provided in our code.

## D.4  DCI-M/C/E

The DCI metrics [18] provide a complementary triad of measures that together offer a comprehensive evaluation of a model's disentanglement capabilities. These metrics assess the ability of a model to produce latent representations $z$ that are both interpretable and structured. The evaluation proceeds by training regressors to predict ground truth generative factors from the learned latent codes. As a joint preprocessing step, let $M$ denote a trained model and $z = \text{Enc}_M(x)$ be the latent representation of input $x$. Let $J = \{j_1, \ldots, j_m\}$ be a partition of the indices of $z$, and $F = \{f_1, \ldots, f_m\}$ the set of $m$ ground truth factors. For each $z_{j_i}$, a regressor $r_{j_i}^{f_k}$ is trained to predict factor $f_k$. In the case where $z$ has a temporal structure (e.g., in time series data), it is reshaped to a fixed-length vector by flattening and averaging across time steps to enable standard regression. After training this $m \times m$ matrix of regressors, the DCI metrics are computed as described in [18]. We use Gradient Boosting Trees as the base regressor throughout our experiments.

## D.5  Consistency metrics for disentanglement

In sequential data, e.g., video, ensuring that the learned representations maintain consistency over time is critical for evaluating disentanglement—especially for static features that should not vary across frames. A disentangled representation should encode the same static factors across a sequence while allowing dynamic factors to vary. We define two primary approaches to measure consistency: i) Swap Consistency: checks preservation of static features after swapping between real examples; and ii) Generation Consistency: evaluates consistency within generated time series, both globally and locally.

**Swap Consistency (C-Swap).**  C-Swap measures whether static features remain temporally consistent when they are transferred between two examples. Let each example $x_i$ be described by static factors $f_{s1}, \ldots, f_{sk}$ and dynamic factors $f_{d1}, \ldots, f_{dn}$, with corresponding factor realizations $v_i = \{s_i, d_i\}$. Here, $s_i \in \mathbb{R}^k$ are static values, and $d_i(t) \in \mathbb{R}^n$ are dynamic values at time step $t \in \{1, \ldots, T\}$. Given two examples $x_1, x_2$, and their static factors $s_1, s_2$, define a subset $f \subseteq \{1, \ldots, k\}$ of static features to be swapped, and let $\overline{f}$ be its complement. We construct swapped examples with the following factor compositions:

$$\tilde{v}_1(t) = (s_2^f, s_1^{\overline{f}}, d_1(t)), \quad \tilde{v}_2(t) = (s_1^f, s_2^{\overline{f}}, d_2(t)) \,. \tag{1}$$

To assess consistency, we compare the label of each swapped feature $m \in f$ over time against its correct value from the source example using a classifier $C_m$. The per feature C-Swap scores are:

$$C_{1\text{-m,swap}} = \frac{1}{T} \sum_{t=1}^{T} \mathbb{I}(C_m(\tilde{x}_1(t)) = v_{2,sm}) \tag{2}$$

$$C_{2\text{-m,swap}} = \frac{1}{T} \sum_{t=1}^{T} \mathbb{I}(C_m(\tilde{x}_2(t)) = v_{1,sm}) \,, \tag{3}$$

here, $v_{2,sm}$ and $v_{1,sm}$ are the original static feature labels from the source examples. These metrics quantify how accurately the swapped features are preserved throughout the sequence.

**Global and Local Generation Consistency (GC-Sample/C-Sample).**  GC-Sample and C-Sample evaluate the internal temporal coherence of static factors in model-generated sequences. Unlike in C-Swap, ground truth static labels are unknown, so we evaluate consistency based on redundancy and stability over time.

GC-Sample assesses if a static feature retains a dominant value across all frames in a generated sequence. Let $\tilde{x}_{\text{gen}}(t)$ be the $t$-th frame of a generated sequence and $C_m(\tilde{x}_{\text{gen}}(t))$ the predicted label of static feature $m$. We first identify the most frequent value of feature $m$ across time:

$$v_{\text{frequent},sm} = \text{mode}\left(C_m(\tilde{x}_{\text{gen}}(1)), \ldots, C_m(\tilde{x}_{\text{gen}}(T))\right) . \tag{4}$$

Then, the GC-Sample score for feature $m$ ranges from 0 to 1 and reflects how dominant the most frequent label is throughout the sequence and it is defined as:

$$C_{\text{m-global}} = \frac{1}{T} \sum_{t=1}^{T} \mathbb{I}(C_m(\tilde{x}_{\text{gen}}(t)) = v_{\text{frequent},sm}) . \tag{5}$$

C-Sample captures short-term continuity by measuring how often static features remain unchanged between consecutive frames. Formally, the C-Sample score for feature $m$ is:

$$C_{\text{m-local}} = \frac{1}{T-1} \sum_{t=1}^{T-1} \mathbb{I}(C_m(\tilde{x}_{\text{gen}}(t)) = C_m(\tilde{x}_{\text{gen}}(t+1))) . \tag{6}$$

This measures the proportion of adjacent frame pairs where the static feature value remains the same. Unlike GC-Sample, C-Sample is sensitive to short bursts of change or fluctuations. Both global and local scores offer complementary views of consistency. High GC-Sample implies that one label dominates across time, even if some changes occur, whereas high C-Sample implies smooth transitions with minimal frame-to-frame variation. In practice, both metrics should be considered when evaluating temporal stability of disentangled static features in generated sequences.

# E    Limitations and future directions

Our benchmark takes a significant step toward providing a flexible, extensible, and scalable framework for the development and evaluation of multi-factor disentanglement methods. Through our benchmarking efforts, we have identified several limitations and promising directions for future research, which we encourage the community to pursue.

**Theoretical grounding.**    Current methods for disentanglement still lack strong theoretical guarantees, particularly in the multi-factor and sequential setting. Most existing approaches assume full statistical independence between factors, despite the fact that real-world generative processes often exhibit causal dependencies [52]. While the field of disentangled representation learning has seen significant theoretical advances in recent years [23, 14], these developments have not yet been adequately extended to the multi-factor case. We believe that bridging this gap by incorporating insights from causal inference and identifiability theory could lay the foundation for more principled models capable of handling complex, structured factor interactions over time.

**Refined factorial swap and sample metrics.**    The four factorial swap and sample metrics used in our benchmark are computed as uniformly weighted arithmetic means of absolute differences between actual and expected accuracies. These values capture two complementary aspects of disentanglement: partition strength (how well each factor is represented by its designated latent subspace) and leakage resistance (how little information about a factor is contained in latent subspaces associated with other factors).

Values representing partition strength are bounded by 1, while values representing leakage resistance have lower upper bounds that depend on the number of classes (the more classes, the higher the bound). However, in multi-factor settings, the number of off-diagonal elements (leakage resistance terms) exceeds the number of on-diagonal elements (partition strength terms). Because all terms are equally weighted, the resulting metric can become dominated by leakage resistance values, despite their lower range. This imbalance increases with the number of factors, leading to metrics that are less robust, less comparable across datasets, and more sensitive to inter-dataset variability. Moreover, the unequal bounds compress the effective range of the final score, biasing it toward higher values.

These metrics can be refined by (1) normalizing each difference to the $[0, 1]$ range, (2) computing separate arithmetic means for partition strength and leakage resistance, and (3) combining them via a weighted geometric mean. This formulation explicitly controls the relative contributions of partition strength and leakage resistance, resulting in more interpretable, balanced, and dataset-independent scores.

**Alternative approaches to sequential disentanglement.**    Current multi-factor sequential disentanglement methods grew in an atmosphere that emphasized an approach of static-dynamic sequential disentanglement and, as a result, they may have inherited some of its pitfalls (e.g., a nuanced and dataset- and modality-dependent definition of static and dynamic factors). Real-world data often does not exhibit a clear perceptual dichotomy between static and dynamic factors and may even exhibit causal relations between them [52] (e.g., the perceived hair color may depend on changes in lighting), thus other approaches are worth exploring, including a multi-scale (possibly hierarchical) approach, which relates sequence dynamics to different time scales of variation - a property of real-world data across all modalities.

**Practical performance.**    Our benchmarking reveals that current methods struggle to generalize to real-world datasets, particularly those involving complex audio and time series data. As illustrated in our failure case analysis, these models often fail to preserve high-level semantic details when manipulating or reconstructing samples. This is partly due to their reliance on VAEs, which are known to produce blurry reconstructions and suffer from posterior collapse in some cases. We argue that future work should explore more expressive generative models - such as diffusion or flow-based [16, 20, 7, 45, 4] models - that may offer better fidelity, robustness, and semantic controllability in the disentanglement setting.

**Leveraging large-scale pretrained models.**    Recent advances in large-scale pretraining have demonstrated that representation learning at scale can yield generalizable and semantically rich latent spaces [32, 60]. While our work does not directly explore these models, we see great potential

in investigating how pre-trained models - especially foundation models in vision, language, and multimodal domains - can be adapted or fine-tuned to produce disentangled representations. This could lead to plug-and-play disentanglement modules, transfer learning across modalities, or domain-specific finetuning with minimal supervision.

**VLMs as taggers, judges, and feedback modules.**   In this work, we take an initial step toward leveraging VLMs for zero-shot annotation and evaluation of semantic factors. This opens new opportunities for replacing or supplementing human supervision. However, further improvements are possible: few-shot learning, adapter-based post-training, and prompt tuning could make VLMs more specialized for disentanglement tasks. Moreover, we envision a future where VLMs not only tag or judge outputs, but actively serve as feedback modules - providing signal for contrastive, reinforcement-based, or hybrid learning objectives to guide unsupervised disentanglement. While such feedback would still be imperfect, it could help bridge the gap between unsupervised objectives and human-aligned semantics.

Overall, we hope our benchmark catalyzes further theoretical, empirical, and practical advancements in the pursuit of robust multi-factor disentangled representations.

# F  LES and VLM additional details

## F.1  LES

We introduce this stage prior to presenting the evaluation metrics, as it is a prerequisite for effectively assessing sequential disentanglement models. Evaluating unsupervised disentanglement is a well-known and ongoing challenge [51]. In the context of sequential models, this issue is particularly pronounced - even when ground truth labels are available - because these methods generally assume a non-compact latent representation and require tedious human intervention. That is, semantic factors (e.g., hair color or age) may be encoded across multiple latent dimensions, rather than being captured by a single variable, as would be expected under the compactness assumption [11]. This makes manual inspection even more difficult and error-prone.

Even when a model is successfully trained, a poor choice of latent-to-factor mapping may severely degrade performance on downstream disentanglement tasks. Tab. 7 illustrates this issue: we compare results on a multi-factor swapping task [6] using the authors' publicly released model, with and without our proposed LES. The results show significant performance gains obtained solely by intelligently selecting the latent-factor mapping, highlighting the importance of post-hoc exploration.

**LES.**   The LES aims to identify which components of a learned latent representation correspond to specific generative factors. Given a pretrained model, LES techniques allow interpretation of the latent space by constructing mappings between dimensions and known factors of variation. The benchmark includes two complementary exploration strategies: *predictor-based* and *swap-based* methods. Both techniques are designed to integrate seamlessly into the benchmark, and users can choose based on their desired speed-quality trade-off. Moreover, LES is designed to be modular: new exploration techniques can be added to the framework with minimal effort, providing flexibility for future extensions.

Table 7: Comparison of SKD without and with LES on Sprites dataset.

|  | SKD w/o LES | SKD + LES |
|---|---|---|
| M-Swap | 0.69 | 0.70 |
| M-GSample | 0.67 | 0.71 |

**Predictor-based LES**   The predictor-based approach evaluates the informativeness of each latent dimension by training a supervised classifier to predict ground truth factors from the latent codes. Specifically, for each factor, we train an independent predictor (e.g., gradient boosting classifier) using the latent representations as input and factor labels as targets. Classification accuracy serves as a proxy for how well the latent space captures the given factor. To localize the influence of specific dimensions, we extract feature importances from each trained predictor, enabling the construction of a mapping from factors to influential latent dimensions. This method is fast and effective when full supervision is available but may suffer when classifiers are inaccurate or overfit.

**Swap-based LES**   The swap-based method assesses disentanglement by applying targeted interventions in the latent space. The idea is to swap a subset of latent dimensions between two samples, decode the resulting representations, and observe which semantic factors change. Given a candidate latent subset, we swap it between two samples and use a pretrained judge model to classify the decoded outputs. If a specific factor changes consistently when a particular latent subset is modified, we infer that this subset encodes the corresponding factor. To promote minimal and disentangled mappings, we penalize large subsets during exploration. While this method can be more precise than the predictor-based approach, it is computationally more intensive due to the need for repeated decoding and classifier evaluation and aims to discover how well all combinatorial options perform, while the predictor-based doesn't consider these combinations.

To ensure fair evaluation, we used predictor-based LES uniformly across all methods and datasets in our reported benchmark.

## F.2  VLM implementation details

We integrate VLMs into our benchmark framework for two primary purposes: (1) automatic dataset annotation and (2) classification of new, previously unseen samples. For both tasks, we employ

OpenAI's GPT-4o via the API as our backbone model; however, the benchmark is compatible with alternative VLM backends, such as Qwen2.5, which is readily integrated in our code.

The automatic annotation tool consists of three key stages (see Fig. 7 for an overview): First in the *feature space discovery* stage (Fig. 8), the VLM is queried with a large number of sample pairs to identify distinguishing attributes (e.g., "blue hair in image 1, red hair in image 2"). Recurrent or semantically similar attributes are grouped to form a concise set of varying factors, with user-defined thresholds controlling factor inclusion (e.g., features that vary in at least 10% of comparisons). Second, in the *label space discovery* stage (Fig. 9), the VLM uses the previously extracted factor descriptions to generate a set of likely discrete label values per factor. Finally, during the *annotation* stage (Fig. 10), the model is presented with each sample and asked to assign the appropriate label for each factor using a closed-set, multiple-choice format.

This same closed-label querying procedure is also employed in the *classifier/judge* module, where the VLM, given a known factor and its predefined label set, selects the most appropriate label for a new sample, thus eliminating the need for dataset-specific classifiers.

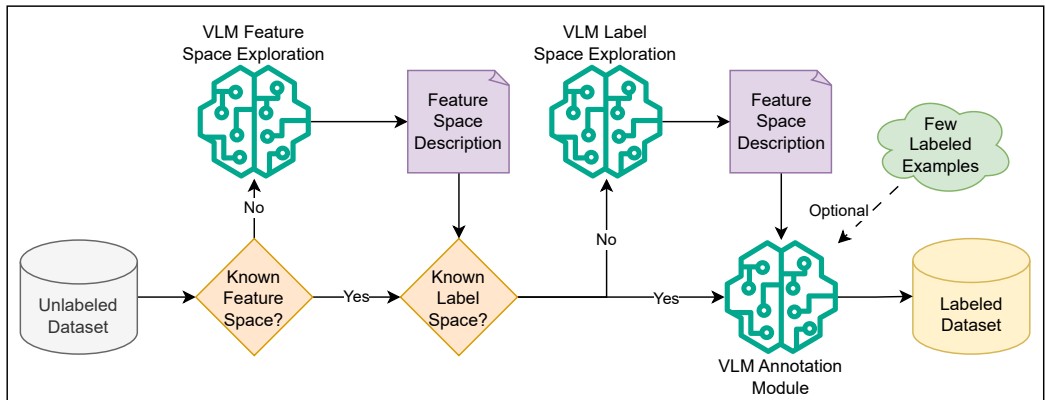

Figure 7: **VLM-based annotation framework.** Overview of our pipeline for automatic dataset annotation using a VLM. Given an unlabeled dataset, the system optionally performs feature space and label space discovery if they are not known in advance. Once both spaces are defined, the VLM assigns labels to each sample through closed-set queries. Few-shot examples can optionally be provided to guide the annotation process. This same procedure is reused for classification of previously unseen samples.

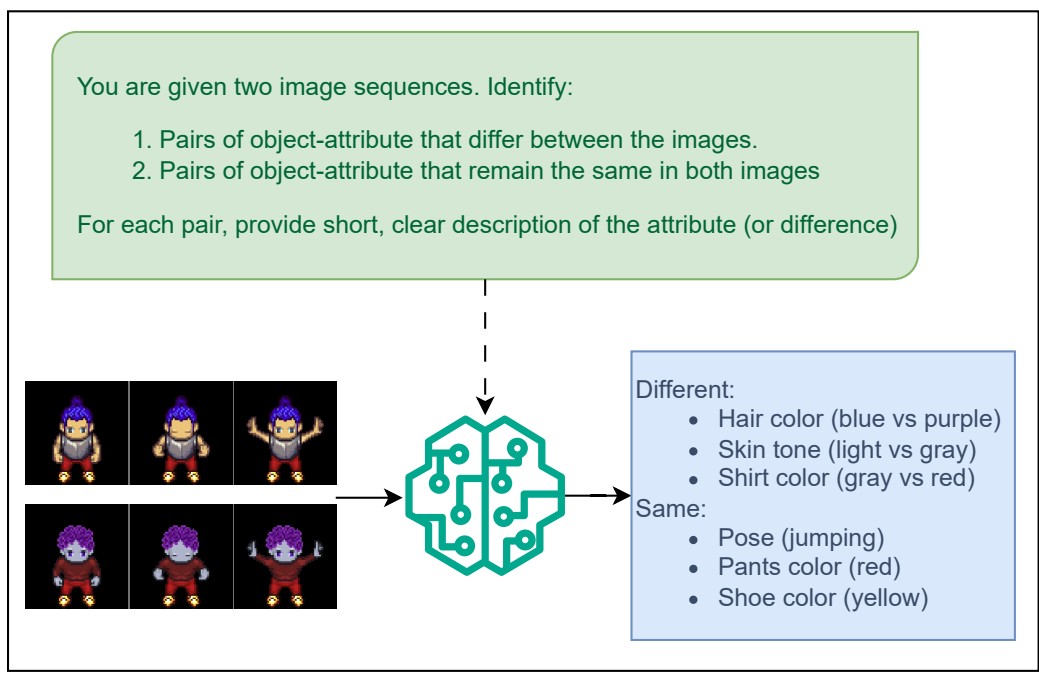

Figure 8: **Comparison-based feature exploration.** The model receives two image sequences and identifies which visual attributes differ or remain the same.

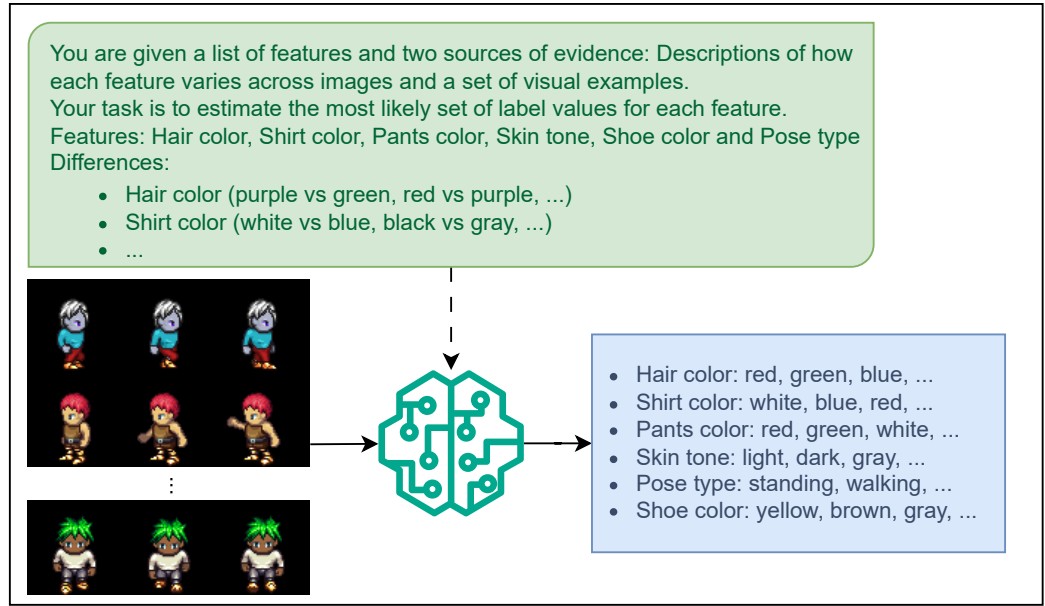

Figure 9: **Label space estimation.** The model uses visual and textual evidence to estimate a practical label set for each known feature.

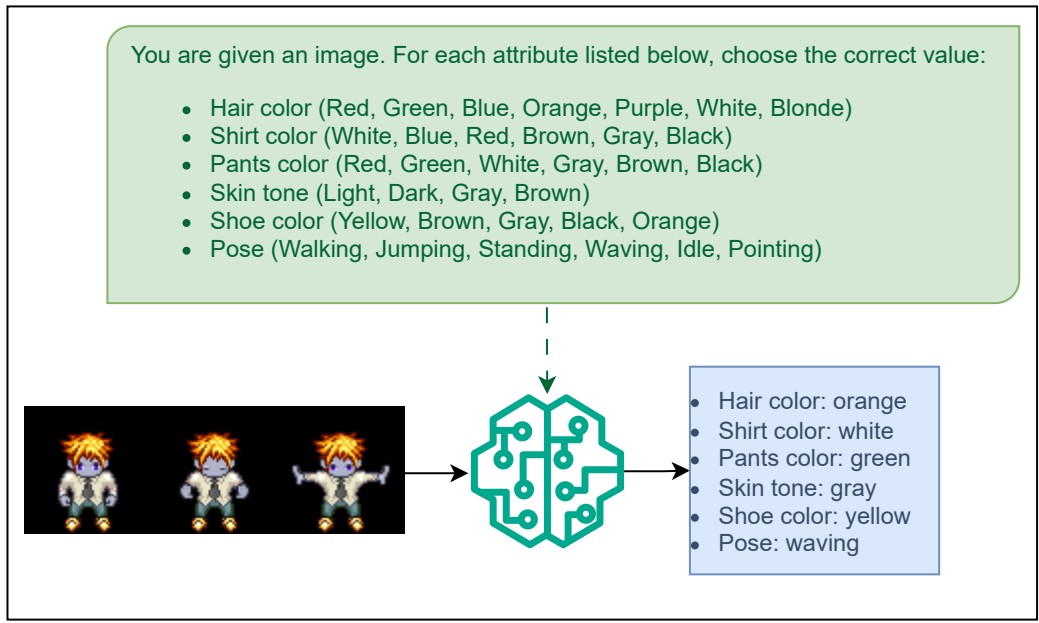

Figure 10: **Annotation/judging.** The model is presented with a single image (or sequence) and a predefined set of attributes. For each attribute, it must select the correct value from a fixed label space. This module is used to annotate unseen samples in zero-shot or few-shot settings.

### F.3 Full VoxCelebOne experiment results

We used the VLM feature and label exploration module to automatically identify semantic factors in the VoxCelebOne dataset (Fig. 11). The following factors, along with their corresponding label spaces, were discovered in an unsupervised manner: background color, hair style, hair color, sex, lighting, and shirt color. A user can manually add factors, and then the Tagger module will find their label space. To expand this set, we manually added additional meaningful factors: age group, skin color, glasses, facial hair, and earrings. The complete list of factors and their possible values is shown in Tab. 9.

Next, we used the VLM annotation module to label 100 samples from the dataset and manually annotated the same set for comparison. We then computed per factor annotation accuracy by comparing VLM predictions against human labels. Results are reported in Tab. 8.

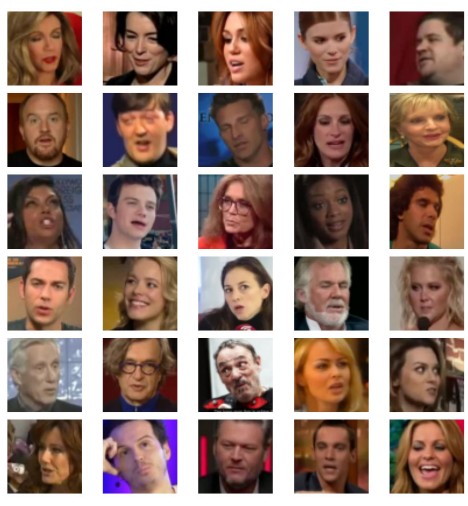

Figure 11: Sampled frames from the VoxCelebOne dataset.

Table 8: Semantic factors with their possible values and VLM annotation accuracy on VoxCelebOne

| Factor | Values | Accuracy |
|---|---|---|
| Background | red, green, ... | 0.92 |
| Hair style | short, wavy, ... | 0.81 |
| Hair color | black, brown, ... | 0.84 |
| Sex | male, female | 1.00 |
| Lighting | bright, dim, ... | 0.72 |
| Shirt color | green, blue, ... | 0.63 |
| Age group | <20, 20–40, ... | 0.65 |
| Skin color | light, medium, ... | 0.89 |
| Glasses | yes, no | 0.99 |
| Facial hair | yes, no | 0.95 |
| Earrings | yes, no | 0.82 |

Table 9: Semantic factors and their possible values

| Factor | Possible Values |
|---|---|
| Background color | red, blue, green, purple, brown, black, white, gray, orange, pink, yellow, beige, teal, multicolored, dark, light |
| Hair style | short, long, wavy, straight, curly, tied up, combed back, layered, tousled, slicked back, bob cut |
| Hair color | black, brown, blonde, gray, red, white, dark, light |
| Sex | male, female |
| Lighting | bright, dark, dim, warm, cool, natural, artificial |
| Shirt color | green, blue, red, pink, orange, white, black, maroon, dark, light |
| Age | <20, 20–40, 40–60, 60–80, 80+ |
| Skin color | light, medium, dark |
| Glasses | yes, no |
| Facial hair | yes, no |
| Earrings | yes, no |

# G Extended results and experimental setup

## G.1 Full comparison per dataset

We provide an extended comparison of all methods across all metrics, broken down by dataset. These detailed tables complement the summary presented in Tab. 1 of the main paper. For each dataset, we present a dedicated table: Sprites (Tab. 10), 3D Shapes (Tab. 11), dSprites-Static (Tab. 12), dSprites-Dynamic (Tab. 13), dMelodies-WAV (Tab. 14), and BMS Air Quality (Tab. 15). All results have an additional level of detail; for simplicity, we attach these full results in the supplementary material files.

Table 10: Performance of disentanglement methods on the **Sprites** dataset. ↑ indicates that higher is better. Bold values denote the best score per row.

| Metric | Sparse-AE | VAE | $\beta$-VAE | MGP-VAE | SKD | SSM-SKD |
|--------|-----------|-----|-------------|---------|-----|---------|
| M-Swap ↑ | 0.69 ± 1e-03 | 0.64 ± 2e-03 | 0.66 ± 2e-03 | 0.86 ± 2e-03 | 0.70 ± 6e-03 | **0.94 ± 3e-03** |
| 2-Swap ↑ | 0.65 ± 3e-03 | 0.63 ± 3e-03 | 0.66 ± 2e-03 | 0.89 ± 3e-03 | 0.97 ± 2e-02 | **0.97 ± 2e-03** |
| M-GSample ↑ | 0.69 ± 2e-03 | 0.65 ± 1e-03 | 0.66 ± 1e-03 | 0.62 ± 3e-03 | 0.71 ± 1e-02 | **0.94 ± 2e-03** |
| 2-GSample ↑ | 0.66 ± 3e-03 | 0.63 ± 3e-03 | 0.66 ± 2e-03 | 0.81 ± 2e-03 | **0.96 ± 2e-02** | 0.96 ± 4e-03 |
| DCI-M ↑ | 0.37 ± 6e-03 | 0.25 ± 4e-03 | 0.28 ± 3e-03 | 0.57 ± 4e-03 | 0.30 ± 5e-03 | **0.89 ± 3e-03** |
| DCI-C ↑ | 0.83 ± 1e-03 | 0.78 ± 1e-03 | 0.79 ± 6e-04 | 0.73 ± 3e-03 | 0.71 ± 4e-03 | **0.95 ± 1e-03** |
| DCI-E ↑ | 0.80 ± 3e-03 | 0.33 ± 9e-03 | 0.33 ± 4e-03 | 0.94 ± 4e-03 | 0.77 ± 2e-03 | **0.98 ± 9e-04** |
| C-Swap ↑ | 0.65 ± 2e-03 | 0.17 ± 7e-04 | 0.17 ± 4e-04 | 0.91 ± 1e-03 | 0.49 ± 2e-02 | **0.95 ± 8e-04** |
| C-Sample ↑ | **0.97 ± 7e-04** | 0.87 ± 2e-03 | 0.88 ± 2e-03 | 0.79 ± 2e-03 | 0.95 ± 5e-04 | 0.96 ± 6e-04 |
| GC-Sample ↑ | **0.97 ± 1e-03** | 0.85 ± 3e-03 | 0.86 ± 3e-03 | 0.76 ± 4e-03 | 0.96 ± 4e-03 | 0.97 ± 4e-04 |

Table 11: Performance of disentanglement methods on the **3D Shapes** dataset. ↑ indicates that higher is better. Bold values denote the best score per row.

| Metric | Sparse-AE | VAE | $\beta$-VAE | MGP-VAE | SKD | SSM-SKD |
|--------|-----------|-----|-------------|---------|-----|---------|
| M-Swap ↑ | 0.79 ± 8e-04 | 0.87 ± 2e-04 | 0.72 ± 8e-04 | 0.76 ± 5e-04 | 0.80 ± 8e-04 | **0.93 ± 6e-04** |
| 2-Swap ↑ | 0.88 ± 1e-03 | 0.93 ± 9e-04 | 0.90 ± 8e-04 | 0.85 ± 1e-03 | **0.98 ± 7e-04** | 0.97 ± 1e-03 |
| M-GSample ↑ | 0.64 ± 6e-04 | 0.86 ± 1e-03 | 0.73 ± 7e-04 | 0.62 ± 5e-04 | 0.81 ± 8e-04 | **0.95 ± 5e-04** |
| 2-GSample ↑ | 0.74 ± 1e-03 | 0.93 ± 8e-04 | 0.91 ± 6e-04 | 0.71 ± 1e-03 | **0.97 ± 7e-04** | 0.97 ± 8e-04 |
| DCI-M ↑ | 0.75 ± 2e-03 | 0.90 ± 2e-03 | 0.54 ± 1e-03 | 0.25 ± 4e-03 | 0.23 ± 2e-03 | **0.92 ± 3e-04** |
| DCI-C ↑ | 0.92 ± 5e-04 | **0.97 ± 6e-04** | 0.86 ± 2e-04 | 0.52 ± 2e-03 | 0.63 ± 3e-03 | 0.97 ± 1e-04 |
| DCI-E ↑ | 0.96 ± 1e-03 | 0.99 ± 7e-04 | 0.92 ± 2e-03 | 0.45 ± 4e-03 | 0.58 ± 3e-04 | **1.00 ± 9e-05** |
| C-Swap ↑ | 0.80 ± 1e-03 | 0.90 ± 3e-04 | 0.63 ± 8e-04 | 0.80 ± 1e-03 | 0.63 ± 2e-03 | **0.95 ± 3e-04** |
| C-Sample ↑ | 0.99 ± 3e-04 | 1.00 ± 2e-04 | **1.00 ± 2e-05** | 0.81 ± 1e-03 | 0.98 ± 1e-04 | 0.98 ± 2e-04 |
| GC-Sample ↑ | 0.99 ± 3e-04 | 1.00 ± 1e-04 | **1.00 ± 1e-04** | 0.79 ± 1e-03 | 0.98 ± 2e-04 | 0.98 ± 4e-05 |

Table 12: Performance of disentanglement methods on the **dSprites-Static** dataset. ↑ indicates that higher is better. Bold values denote the best score per row.

| Metric | Sparse-AE | VAE | $\beta$-VAE | MGP-VAE | SKD | SSM-SKD |
|---|---|---|---|---|---|---|
| M-Swap ↑ | 0.60 ± 7e-04 | 0.60 ± 4e-04 | 0.64 ± 2e-03 | 0.56 ± 4e-04 | 0.65 ± 1e-03 | **0.78 ± 2e-03** |
| 2-Swap ↑ | 0.60 ± 1e-03 | 0.60 ± 3e-04 | 0.77 ± 6e-04 | 0.72 ± 2e-03 | 0.77 ± 2e-03 | **0.80 ± 5e-03** |
| M-GSample ↑ | 0.60 ± 1e-03 | 0.60 ± 4e-04 | 0.65 ± 1e-03 | 0.49 ± 7e-04 | 0.66 ± 1e-03 | **0.79 ± 3e-03** |
| 2-GSample ↑ | 0.60 ± 8e-04 | 0.60 ± 7e-04 | 0.78 ± 4e-04 | 0.65 ± 1e-03 | 0.78 ± 1e-03 | **0.81 ± 3e-03** |
| DCI-M ↑ | 0.00 ± 4e-04 | 0.00 ± 3e-04 | 0.23 ± 3e-03 | 0.09 ± 3e-03 | 0.09 ± 6e-04 | **0.69 ± 2e-03** |
| DCI-C ↑ | 0.68 ± 3e-04 | 0.68 ± 7e-04 | 0.76 ± 1e-03 | 0.44 ± 3e-03 | 0.57 ± 2e-03 | **0.86 ± 9e-04** |
| DCI-E ↑ | 0.20 ± 6e-03 | 0.21 ± 8e-03 | 0.69 ± 9e-03 | 0.34 ± 2e-03 | 0.58 ± 9e-04 | **0.95 ± 3e-04** |
| C-Swap ↑ | 0.17 ± 3e-04 | 0.17 ± 3e-04 | 0.55 ± 2e-03 | 0.61 ± 1e-03 | 0.50 ± 1e-03 | **0.75 ± 5e-04** |
| C-Sample ↑ | 0.75 ± 9e-04 | 0.85 ± 5e-04 | **0.98 ± 3e-04** | 0.72 ± 2e-03 | 0.91 ± 5e-04 | 0.94 ± 7e-04 |
| GC-Sample ↑ | 0.81 ± 4e-04 | 0.92 ± 2e-03 | **0.99 ± 2e-04** | 0.71 ± 1e-03 | 0.93 ± 3e-04 | 0.95 ± 8e-04 |

Table 13: Performance of disentanglement methods on the **dSprites-Dynamic** dataset. ↑ indicates that higher is better. Bold values denote the best score per row.

| Metric | Sparse-AE | VAE | $\beta$-VAE | MGP-VAE | SKD | SSM-SKD |
|---|---|---|---|---|---|---|
| M-Swap ↑ | 0.65 ± 2e-03 | 0.62 ± 2e-03 | 0.65 ± 1e-03 | **0.70 ± 6e-04** | 0.67 ± 3e-03 | 0.67 ± 7e-04 |
| 2-Swap ↑ | 0.68 ± 3e-03 | 0.65 ± 1e-03 | 0.59 ± 3e-03 | 0.70 ± 2e-03 | **0.77 ± 2e-03** | 0.70 ± 2e-03 |
| M-GSample ↑ | 0.61 ± 9e-04 | 0.62 ± 6e-04 | 0.66 ± 7e-04 | 0.60 ± 8e-04 | 0.68 ± 7e-03 | **0.69 ± 2e-03** |
| 2-GSample ↑ | 0.62 ± 1e-03 | 0.66 ± 1e-03 | 0.59 ± 3e-03 | 0.60 ± 1e-03 | **0.77 ± 2e-03** | 0.69 ± 2e-03 |
| DCI-M ↑ | 0.28 ± 3e-03 | 0.14 ± 1e-03 | 0.20 ± 3e-03 | 0.10 ± 3e-03 | 0.08 ± 1e-03 | **0.34 ± 2e-03** |
| DCI-C ↑ | 0.77 ± 1e-03 | 0.73 ± 5e-04 | **0.78 ± 5e-04** | 0.44 ± 2e-03 | 0.57 ± 4e-03 | 0.72 ± 2e-03 |
| DCI-E ↑ | 0.68 ± 4e-03 | 0.59 ± 3e-03 | 0.54 ± 8e-03 | 0.35 ± 4e-03 | 0.55 ± 2e-03 | **0.77 ± 2e-03** |
| C-Swap ↑ | 0.59 ± 1e-03 | 0.48 ± 2e-03 | 0.62 ± 1e-03 | 0.33 ± 9e-04 | 0.48 ± 1e-02 | **0.67 ± 1e-03** |
| C-Sample ↑ | **0.94 ± 4e-04** | 0.89 ± 2e-03 | 0.94 ± 7e-04 | 0.74 ± 1e-03 | 0.90 ± 8e-04 | 0.93 ± 6e-04 |
| GC-Sample ↑ | 0.95 ± 7e-04 | 0.94 ± 5e-04 | **0.96 ± 3e-04** | 0.72 ± 7e-04 | 0.91 ± 7e-04 | 0.94 ± 1e-03 |

Table 14: Performance of disentanglement methods on the **dMelodies-WAV** dataset. ↑ indicates that higher is better. Bold values denote the best score per row.

| Metric | Sparse-AE | VAE | $\beta$-VAE | MGP-VAE | SKD | SSM-SKD |
|---|---|---|---|---|---|---|
| M-Swap ↑ | 0.63 ± 0e+00 | 0.63 ± 0e+00 | 0.63 ± 0e+00 | 0.53 ± 0e+00 | 0.63 ± 0e+00 | **0.65 ± 2e-03** |
| M-GSample ↑ | 0.63 ± 0e+00 | 0.63 ± 0e+00 | 0.63 ± 0e+00 | 0.53 ± 0e+00 | 0.63 ± 0e+00 | **0.65 ± 2e-03** |
| DCI-M ↑ | 0.02 ± 1e-03 | 0.01 ± 2e-03 | **0.17 ± 2e-03** | 0.00 ± 8e-04 | 0.09 ± 5e-03 | 0.13 ± 3e-03 |
| DCI-C ↑ | 0.68 ± 7e-04 | 0.69 ± 2e-03 | **0.75 ± 2e-03** | 0.39 ± 5e-03 | 0.62 ± 4e-03 | 0.67 ± 2e-03 |
| DCI-E ↑ | 0.38 ± 1e-02 | 0.29 ± 4e-03 | 0.41 ± 8e-03 | 0.29 ± 8e-03 | 0.64 ± 2e-03 | **0.67 ± 2e-03** |

Table 15: Performance of disentanglement methods on the **BMS Air Quality** dataset. ↑ indicates that higher is better. Bold values denote the best score per row.

| Metric | Sparse-AE | VAE | $\beta$-VAE | MGP-VAE | SKD | SSM-SKD |
|---|---|---|---|---|---|---|
| DCI-M ↑ | 0.06 ± 2e-03 | **0.16 ± 2e-03** | 0.16 ± 3e-03 | 0.00 ± 7e-04 | 0.07 ± 3e-03 | 0.12 ± 2e-03 |
| DCI-C ↑ | 0.74 ± 1e-03 | 0.78 ± 7e-04 | **0.78 ± 2e-03** | 0.33 ± 2e-03 | 0.68 ± 2e-02 | 0.70 ± 6e-04 |
| DCI-E ↑ | 0.29 ± 1e-02 | 0.32 ± 5e-03 | 0.31 ± 6e-03 | 0.17 ± 4e-03 | 0.33 ± 8e-03 | **0.43 ± 3e-03** |

### G.2 Full metric results

Certain evaluation metrics presented in the main paper were distilled into a single representative score for simplicity and clarity. Here, we provide the complete, undistilled metric results for each dataset. Due to space limitations, these detailed tables are not included directly in the main document but in the supplementary material files.

### G.3 Results with VLM

In Tab. 16 and Tab. 17 we present the final scores obtained using the VLM evaluation. Although these scores differ from those obtained under the setup with full access to ground truth labels, we demonstrate in Sec. 4.4 that the ranking order remains almost perfectly correlated.

Table 16: Performance of disentanglement methods on the **Sprites (VLM)** dataset. ↑ indicates that higher is better. Bold values denote the best score per row.

| **Metric** | Sparse-AE | VAE | $\beta$-VAE | MGP-VAE | SKD | SSM-SKD |
|---|---|---|---|---|---|---|
| M-Swap ↑ | 0.64 | 0.60 | 0.61 | 0.71 | 0.64 | **0.76** |
| DCI-M ↑ | 0.17 | 0.07 | 0.07 | 0.29 | 0.14 | **0.43** |
| DCI-C ↑ | **0.73** | 0.70 | 0.70 | 0.49 | 0.59 | 0.71 |
| DCI-E ↑ | 0.59 | 0.39 | 0.40 | 0.77 | 0.71 | **0.78** |

Table 17: Performance of disentanglement methods on the **3D Shapes (VLM)** dataset. ↑ indicates that higher is better. Bold values denote the best score per row.

| **Metric** | Sparse-AE | VAE | $\beta$-VAE | MGP-VAE | SKD | SSM-SKD |
|---|---|---|---|---|---|---|
| M-Swap ↑ | 0.67 | 0.73 | 0.69 | 0.54 | 0.63 | **0.74** |
| DCI-M ↑ | 0.24 | 0.43 | 0.14 | 0.05 | 0.34 | **0.49** |
| DCI-C ↑ | 0.81 | **0.86** | 0.79 | 0.63 | 0.76 | 0.82 |
| DCI-E ↑ | 0.67 | 0.78 | 0.61 | 0.45 | 0.76 | **0.83** |

### G.4 Experimental setup and computational cost

We produced the above results through training and evaluation on a cluster of machines with AMD EPYC 7002 Series CPU (x86_64 architecture), 256 GB RAM, Linux kernel 5.14.0, glibc 2.34, NVIDIA GeForce RTX 4090 (Gigabyte) GPU, NVIDIA VBIOS 95.02.3C.C0.93, NVIDIA driver 565.57.01, CUDA 12.6, cuDNN 9.5.1, Python 3.9.21, pip 25.1, NumPy 1.26.4, PyTorch 2.7.0, and the packages listed in *requirements.txt* (available in our code).

Training times varied per model and was depending on the dataset, between 6 to 60 seconds for an epoch. Evaluation runs required under 2 hours per method. Preliminary and ablation experiments approximately doubled the total compute usage.

