# OpenReview forum: "Disentanglement Beyond Static vs. Dynamic: A Benchmark and Evaluation Framework for Multi-Factor Sequential Representations"
_NeurIPS.cc/2025/Datasets_and_Benchmarks_Track — NeurIPS 2025 Datasets and Benchmarks Track poster_

### Official Review · Reviewer_KVH3 · 2025-07-01

**Rating:** 5
**Confidence:** 4

**Summary:**

This paper proposes a benchmark for multi-factor sequential disentanglement (MSD), addressing the limitations of prior static/dynamic two-factor frameworks. The benchmark includes: six diverse datasets (video, audio, time-series), ten metrics (adapted or proposed), Vision-language models (VLMs) for zero-shot annotation and evaluation. Authors also have their own methods joined in the comparison list and analyze its failure modes on some particular cases. The latent exploration stage help us investigates the ambiguity during latent-label matching process, which is important for latent disentanglement.

**Dataset Code Accessibility:**

Yes

**Ethical Comments:**

/

**Ethical Considerations:**

No, there are no or only very minor ethics concerns

**Final Justification:**

I will keep my score of 5 (accept) for this dataset paper. Generally speaking, the benchmarking experiments are comprehensive, and I think it is a good dataset/benchmark paper ready to be published in NeurIPS after minor revisions on the clarity/presentation issue.

**Limitations Weaknesses:**

* Some tiny writing issues:
    * line 146 "with features in $\mathbb{R}^{T\times o}$ is a bit confusing
    * line 173 "(2) output a single 1D latent vector for each sample"
* In terms of presentation, the 10 metrics are good, but are not clearly presented, which took me some time but only made me partially understand how it does.
* I'm curious about the validity of some of the metrics. How to visually understand that they are trustable? For some datasets that all methods performs not perfect, is it because of the the dataset itself is challenging or the metrics are not appropriate for evaluation?

**Strengths Contributions:**

* The multi-factor disentanglement is novel and a more general case compared with two two-factor in the field of sequence disentanglement
* The overall work presented in this benchmark paper is comprehensive, encompassing workflow, datasets, methods, evaluations, and manual investigations.
* APIs are open and constructive, which provides a good platform for others to contribute.

---

> ### Author Rebuttal · Authors · 2025-07-30
>
> We thank the reviewer for the constructive and encouraging feedback. We appreciate the recognition of our work’s novelty, comprehensiveness, and practical value. We will revise the noted writing issues and improve the clarity and interpretability of the evaluation metrics as suggested. We’re happy to incorporate all feedback in the final version and provide further clarifications if needed.
>
> ---
>
> > ### **Comment:**
> > Some tiny writing issues:
> > - Line 146: "with features in $\mathcal{R}^{T\times o}$" is a bit confusing
> > - Line 173: "(2) output a single 1D latent vector for each sample"
>
> ### **Response:**
> We appreciate the reviewer’s close reading and helpful suggestions.
>
> **Regarding Line 146:**
> We agree that the notation $\mathbb{R}^{T \times o}$ may be ambiguous, particularly because $o$ can represent either flat or structured feature dimensions. We have clarified the text as follows:
> *"...each sample $x \in X$ is a sequence of length $T$, where each element $x_t$ lies in $\mathbb{R}^o$. For example, in a multivariate time series, $o$ is simply the number of features per time step; in video, $o = c \times h \times w$ denotes a structured image tensor."*
> This revision makes the feature space interpretation more concrete and accessible across modalities.
>
> **Regarding Line 173:**
> The phrase "a single 1D latent vector" refers to the requirement that each method provides a one-dimensional, fixed-sized, latent representation for a full sequence. This can be achieved by flattening, pooling, or applying another temporal summarization strategy over the latent sequence. We have revised the text to:
> *"(2) output a fixed-length latent representation (e.g., a 1D vector) for each input sequence, obtained by flattening or aggregating the full latent sequence."*
> This rephrasing improves clarity while preserving the intent of the benchmark interface.
>
> We thank the reviewer for these helpful comments that led to improved clarity in our manuscript.
>
> ---
>
> > ### **Comment:**
> > In terms of presentation, the 10 metrics are good, but are not clearly presented, which took me some time but only made me partially understand how it does.
>
> ### **Response:**
> We thank the reviewer for raising this important point. In response, we will revise **Section 3.4** to provide a more structured and intuitive overview of the ten evaluation metrics used in our benchmark, and will enhance **Appendix D** with clearer headings and illustrative examples.
>
> The metrics fall into three main categories, each targeting a distinct aspect of multi-factor disentanglement:
>
> - **Factor Swap and Sample Metrics** -- These metrics evaluate how specific latent subspaces control semantic attributes in the output:
>   - **2-Swap** and **2-GSample** assess whether a model can separate static and dynamic factors in a two-factor setup, by swapping or sampling latent components and verifying the result using classifiers or VLMs.
>   - **M-Swap** and **M-GSample** extend this to the multi-factor setting, where the model is expected to modify or preserve individual factors selectively. The metrics summarize intervention matrices into a single score that combines preservation and controlled variation.
>
> - **DCI Metrics (DCI-M, DCI-C, DCI-E)** -- These modality-agnostic metrics evaluate the structure of the learned latent space via supervised prediction:
>   - **DCI-M (Modularity)** measures whether each latent dimension affects only a single factor.
>   - **DCI-C (Compactness)** assesses whether each factor is controlled by a small subset of latents.
>   - **DCI-E (Informativeness)** quantifies how well latent representations retain useful information about the factors.
>
> - **Consistency Metrics (C-Swap, C-Sample, GC-Sample)** -- These are video-specific and evaluate the temporal stability of static factors:
>   - **C-Swap** measures whether static attributes remain consistent when swapped between two video sequences.
>   - **C-Sample** checks whether sampling static factors results in temporally stable outputs.
>   - **GC-Sample** evaluates global consistency by checking whether generated sequences preserve static attributes across time and samples.
>
> We hope these revisions will make the benchmark easier to understand and navigate and we would be more than happy to clarify any further details.
>
> ---
>
> > ### **Comment:**
> > I'm curious about the validity of some of the metrics. How to visually understand that they are trustable? For some datasets that all methods performs not perfect, is it because of the dataset itself is challenging or the metrics are not appropriate for evaluation?
>
> ### **Response:**
> We appreciate the reviewer’s question regarding how to assess the trustworthiness of our evaluation metrics.
>
> Some metrics in our suite are domain-specific by design. For instance, the consistency-based metrics (**C-Swap**, **C-Sample**, **GC-Sample**) are applicable only to video data, where temporal coherence can be meaningfully assessed. Other metrics, such as **M-Swap** and **M-GSample**, are applicable across domains but rely on the availability of accurate per-factor classifiers or judges (e.g., VLMs or pretrained models), which may vary in quality depending on the data modality and label complexity.
>
> To support qualitative evaluation and visual validation, our benchmark includes reconstructor and generator modules that produce outputs for swapped or manipulated latents (on both video and audio data). These allow users to inspect the effect of latent interventions via videos or audio samples, helping build trust in the metrics by aligning numerical results with human intuition.
>
> As observed in our experiments (**Section 4.3** and **Appendix E**), most current models -- including our own -- perform substantially better on clean, synthetic video datasets and struggle to disentangle high-level semantics in real-world domains (e.g., audio, time series). These results reflect limitations in current disentanglement approaches rather than in the evaluation metrics themselves, and we hope our benchmark will serve as a foundation to drive progress in these more challenging domains.

---

> > ### Comment · Reviewer_KVH3 · 2025-08-03
> >
> > I thank the authors for providing these further explanations. The rebuttal solves most of my questions, especially in terms of the clarity issue. I would encourage the authors to include these explanations in the Appendix in the later revisions, and I'll keep my supporting score for this paper.

---

> > > ### Author Response · Authors · 2025-08-06
> > > **Thank You**
> > >
> > > Thank you for engaging in the discussion, which helped us refine our work. We will incorporate the suggested clarification.

---

### Official Review · Reviewer_NcTT · 2025-07-02

**Ethics Flags:** Safety and security
**Rating:** 5
**Confidence:** 5

**Summary:**

A posthoc benchmark is proposed to measure multi-factor disentanglement across modalities. The work offers a unified framework with diverse datasets and automated tools for scalable and reproducible evaluation.

**Dataset Code Accessibility:**

Yes

**Ethical Considerations:**

No, there are no or only very minor ethics concerns

**Final Justification:**

I have read the authors' rebuttal, which addressed most of my concerns during the discussion stage. The remaining issues, such as configuration details and model checkpoints, are expected to be resolved in the released code repository. Therefore, I have decided to increase my score from 3 to 5.

**Limitations Weaknesses:**

* -- I have concerns regarding the evaluation metrics. Numerous disentanglement metrics have been proposed in recent years—such as FactorVAE Score, DCI, MIG, SAP—and many unsupervised scores have been integrated in public toolkits like [Disent](https://github.com/nmichlo/disent). However, results may vary significantly depending on the specific implementation and configuration. More details should be provided on how each metric is configured and selected for evaluation.
* -- According to Table 1, the vanilla VAE achieves the best score on certain datasets. This raises the concern that the benchmark may be biased toward vision data or simplified settings, potentially limiting its meaningfulness for other modalities such as audio or time series.
* -- It is unclear how the final score in Table 1 is computed. A clearer explanation—preferably in the form of an equation—of how different metrics contribute to the overall score would greatly enhance the transparency and interpretability of the evaluation.
* -- In addition, more checkpoints (ckpts) should be released for reproducibility. Since VAE-based methods vary significantly depending on architectural choices and hyperparameter settings, it is important to provide multiple trained variants for robust comparisons.

**Strengths Contributions:**

* ++ The proposed benchmark is cross-modality and integrates a wide range of existing metrics (e.g., DCI family) along with Vision-Language Models (VLMs) for zero-shot evaluation, enabling automated and scalable assessment.
* ++ A novel model, SSM-SKD, is introduced, improving disentanglement quality via enhanced Koopman-based latent decomposition.
* ++ The framework is extended to handle real-world datasets, demonstrating its practical applicability beyond synthetic data.

---

> ### Author Rebuttal · Authors · 2025-07-30
>
> We thank the reviewer for the detailed and constructive feedback. We greatly appreciate the recognition of the diversity, scalability, novelty, and practicality of our work. In the responses below, we have addressed all concerns raised and would be more than happy to provide any further clarification if needed. Given the opportunity, we will incorporate all suggested revisions to fully address the feedback.
>
> ---
>
> > ### **Comment:**
> > *I have concerns regarding the evaluation metrics. Numerous disentanglement metrics have been proposed in recent years—such as FactorVAE Score, DCI, MIG, SAP—and many unsupervised scores have been integrated in public toolkits like Disent. However, results may vary significantly depending on the specific implementation and configuration. More details should be provided on how each metric is configured and selected for evaluation.*
>
> ### **Response:**
> Our benchmark includes two main types of evaluation metrics. The first group comprises the **DCI metrics** (metrics 5–7), which we selected for adaptation because they jointly quantify the three foundational properties of disentangled representations: *Disentanglement*, *Completeness*, and *Informativeness*. These metrics are general-purpose, domain-agnostic, and thus applied to all datasets in the benchmark.
>
> The second group consists of **intervention- and consistency-based metrics** (1–4 and 8–10), which operate through latent manipulations and are used selectively depending on the modality and interpretability of ground-truth factors. For example, **M-Swap** and **M-GSample** extend the classic 2-Swap and 2-GSample metrics from two-factor settings to general multi-factor scenarios. These metrics are used on video and audio datasets where interventions over semantic factors produce well-defined, interpretable outputs (e.g., changing a musical chord). In contrast, such manipulations are harder to interpret in time-series data (e.g., swapping the “month” in weather records), so we limit those datasets to DCI metrics. Finally, consistency-based metrics (C-Swap, C-Sample, GC-Sample) rely on visual-temporal similarity and are applied only to visual datasets.
>
> **DCI configuration:**
> We adapted the DCI metrics from static to sequential multi-factor settings, drawing inspiration from toolkits such as `disent` and  `disentanglement_lib`. Specifically, we flatten the latent sequences to form static representations and compute set-level disentanglement scores per semantic factor, rather than using the original per-dimension formulation. Across all datasets, we use `GradientBoostingClassifier` from `scikit-learn` with default hyperparameters, modifying only `max_depth=3` to mitigate overfitting.
>
> **Other metrics:**
> The remaining metrics -- 2-/M-Swap, 2-/M-GSample, C-Swap, C-Sample, and GC-Sample -- do not require hyperparameter tuning at inference time. They are computed using either zero-shot classification via vision-language models (VLMs) or pre-trained dataset-specific classifiers, which we provide. To support reproducibility and customization, we include all classifier training code and associated hyperparameters in our public repository.
>
> We will include this clarification in the final version of the paper and cite relevant prior work, including `disent` and `disentanglement_lib`, to situate our metric adaptations within the broader literature. Finally, the inclusion of additional metrics, such as the FactorVAE Score, MIG, and SAP, is deferred to future work. We would be happy to provide any further clarification if needed.
>
> ---
>
> > ### **Comment:**
> > *According to Table 1, the vanilla VAE achieves the best score on certain datasets. This raises the concern that the benchmark may be biased toward vision data or simplified settings, potentially limiting its meaningfulness for other modalities such as audio or time series.*
>
> ### **Response:**
>
> We believe this observation reflects a bias in current methods rather than in the benchmark design. In most cases where the VAE outperforms, all models -- including more advanced ones -- struggle, leading to low overall scores. Our qualitative analysis (Fig. 2 in the paper) supports this, confirming that the gap stems from model limitations, not evaluation artifacts. There are also cases, such as `3DShapes`, where VAE meaningfully disentangles factors, as verified by both metrics and inspection. We will include this discussion, along with additional qualitative results on audio and time series data that further illustrate the limitations of both VAE and stronger models, in the final version.
>
> This trend likely results from the field’s historical focus on visual datasets, which are easier to collect and interpret, while audio and time series models are lagging behind. To broaden the scope of disentanglement research, we introduce `dmelodies-wav` (realistic audio) and `BMS Air Quality` (time series) as challenging, structured benchmarks from underrepresented modalities. We hope these additions will serve as catalysts for advancing methods beyond the visual domain.
>
> We will incorporate this discussion into the final version of the paper to emphasize the importance of modality-agnostic approaches and encourage contributions that expand the benchmark to cover additional domains.
>
>
>
> ---
>
> > ### **Comment:**
> > *It is unclear how the final score in Table 1 is computed. A clearer explanation—preferably in the form of an equation—of how different metrics contribute to the overall score would greatly enhance the transparency and interpretability of the evaluation.*
>
> ### **Response:**
>
> Thank you for raising this point and giving us the opportunity to significantly improve the clarity of our work. Each score in Table 1 (in the main paper) corresponds to an aggregated value for a specific (model, dataset) pair, computed as the unweighted average of all applicable normalized metric scores:
>
> $$S_{m,d} = \frac{1}{K_d} \sum_{k=1}^{K_d} s_{m,d,k}$$
>
>
> Where:
> - $S_{m,d}$ is the aggregated score for model $m$ on dataset $d$,
> - $s_{m,d,k} \in [0, 1]$ is the normalized score of model $m$ on metric $k$ for dataset $d$,
> - $K_d$ is the number of applicable metrics for dataset $d$.
>
> This aggregation ensures that each applicable metric contributes equally to the overall score without relying on hand-tuned weights. The result is a single, interpretable scalar per (model, dataset) entry in Table 1.
>
> Following the suggestions by you and another reviewer (sukC), we also added a new **leaderboard table** to expose metric-specific behavior. It presents each model's average performance on each metric across all applicable datasets:
>
> $$L_{m,k} = \frac{1}{|D_k|} \sum_{d \in D_k} s_{m,d,k}$$
>
> Where:
> - $L_{m,k}$ is the leaderboard score of model $m$ on metric $k$,
> - $\mathcal{D}_k$ is the set of datasets for which metric $k$ is applicable.
>
> This leaderboard highlights which models perform best under specific evaluation criteria (e.g., disentanglement, consistency) and provides insight into trade-offs across metrics. Due to space constraints, we refer to the table provided in the sukC rebuttal box.
>
> ---
>
> > ### **Comment:**
> > *In addition, more checkpoints (ckpts) should be released for reproducibility. Since VAE-based methods vary significantly depending on architectural choices and hyperparameter settings, it is important to provide multiple trained variants for robust comparisons.*
>
> ### **Response:**
>
> Thank you for raising this point. To address your concern, we have trained additional checkpoints for each method across all datasets, resulting in a total of **522 new model checkpoints**, as detailed in the table below. For each model, we selected the most impactful hyperparameter configurations based on our hands-on experience and empirical insights. We would be happy to include additional configurations if needed.
>
> Subject to the conference policy (which currently restricts changes to external repositories during the review period), we will upload all checkpoints to the **Hugging Face Hub** for public access once permitted.
>
> > Parameter value sets for each model
>
> | **Model**         | **Parameter**         | **Value Set**                                      |
> |-------------------|------------------------|----------------------------------------------------|
> | **ssm_skd**        | `k_dim`               | {16, 24, 32, 40, 48, 56}                           |
> |                   | `hidden_dim`          | {80, 120, 160, 200}                                |
> | **skd**            | `k_dim`               | {26, 40, 54}                                       |
> |                   | `hidden_dim`          | {80, 90, 110, 140, 180}                            |
> | **mgp_vae**        | `fac`                 | {0.1, 0.5, 0.9}                                    |
> |                   | `kl_beta`             | {2.0, 3.0}                                         |
> | **vae**            | `latent_dim`          | {64, 128, 256}                                     |
> |                   | `hidden_dims`         | {(32, 64, 128, 256), (64, 128, 256, 512)}          |
> | **beta_vae**       | `latent_dim`          | {64, 128, 256}                                     |
> |                   | `beta`                | {2, 3, 5}                                          |
> |                   | `hidden_dims`         | {(32, 64, 128, 256), (64, 128, 256, 512)}          |
> | **sparse_ae**      | `latent_dim`          | {64, 128, 25}                                      |
> |                   | `sparsity_weight`     | {0.01, 0.1, 1.0}                                   |
> |                   | `hidden_dims`         | {(32, 64, 128, 256), (64, 128, 256, 512)}          |
>
> We will also include in our revised paper a comprehensive report of the full grid search originally conducted, which explored a broader range of parameter values.

---

> > ### Comment · Reviewer_NcTT · 2025-08-04
> >
> > Thanks for your rebuttal, I will increase my score.

---

> > > ### Author Response · Authors · 2025-08-06
> > > **Thank You**
> > >
> > > Thank you very much for your valuable feedbacks.

---

### Official Review · Reviewer_53Qu · 2025-07-03

**Rating:** 5
**Confidence:** 3

**Summary:**

This paper proposes a unified benchmark for multi-factor sequential disentanglement (MSD), which addresses the limitations of traditional sequential disentanglement settings that mostly assume a binary static/dynamic factorization. The authors construct a diverse benchmark suite spanning three modalities (video, audio, and time-series) based on 7 carefully reformatted datasets. These datasets share a standardized interface and contain semantically rich factors beyond binary labels, such as gender, shirt color, or motion speed, allowing for a more realistic and fine-grained evaluation of disentanglement.
To overcome the challenges of evaluating disentanglement models at scale, the paper introduces an automation pipeline using VLMs. VLMs are employed both to tag semantic factors and to judge the effects of latent interventions, enabling fully automatic, annotation-free evaluation even on real-world datasets.

**Dataset Code Accessibility:**

Yes

**Dataset Code Comments:**

Their github repo is provided and its well-documented with full instruction from installation to running training and evaluation: https://github.com/azencot-group/MSD-Benchmark. It also includes detailed descriptions of how to run each stage of the pipeline (e.g. Latent Exploration, etc.)

**Ethical Considerations:**

No, there are no or only very minor ethics concerns

**Final Justification:**

My concerns were clarified by the authors' response, and I raise my score to 5 because I believe the method is scalable and useful for future research (e.g. interpretability).

**Limitations Weaknesses:**

- The proposed method SSM-SKD (Sec. 3.6) is quite relevant to MSP: https://arxiv.org/abs/2210.05972 (and subsequent works  https://arxiv.org/abs/2305.18484, https://arxiv.org/abs/2405.19296). They solve the least square problem in latent space (Fig.2) to infer sequence-wise action matrix. They also show the features are disentangled (Fig 7). Can you clarify the difference between the SSM SKD and MSP?

**Strengths Contributions:**

- Scalable framework to assess the disentanglement quality enabled by the recent Multimodal LLMs. The authors also verify the LLMs evaluation alignment to ground truth evaluation.
- The work introduces Latent Exploration Stage (LES), which is a framework to match the latent variables to semantic factors, which has traditionally been done manually. The authors introduce two scalable strategies (prediction based and swap-based)
- They also show clear failure cases on real world data (e.g., VoxCelebOne), highlighting where existing models (including their own) break down, and motivating future work on more expressive generative architectures (e.g., diffusion models or GANs).
- They present a comprehensive set of 10 disentanglement metrics, including improved versions of existing ones and novel consistency-based scores designed specifically for video data.

---

> ### Author Rebuttal · Authors · 2025-07-30
>
> Thank you for the thoughtful and constructive review, and for recognizing the scalability, broad applicability, and transparency of our benchmark. We appreciate your insights, particularly regarding the connection between SSM-SKD and MSP. We address your concerns in detail below, and will incorporate the suggested clarifications and related work discussion into the final version of the paper, given the opportunity.
>
> ---
>
> > ### **Comment:**
> > *The proposed method SSM-SKD (Sec. 3.6) is quite relevant to MSP: <https://arxiv.org/abs/2210.05972> (and subsequent works <https://arxiv.org/abs/2305.18484>, <https://arxiv.org/abs/2405.19296>). They solve the least square problem in latent space (Fig. 2) to infer sequence-wise action matrix. They also show the features are disentangled (Fig. 7). Can you clarify the difference between the SSM-SKD and MSP?*
>
> ### **Response:**
> We agree that MSP offers a compelling approach for uncovering symmetries in data through representation learning. MSP models each sequence using a linear operator **M**, similar in spirit to Dynamic Mode Decomposition (DMD), where **M** predicts future latent representations from past ones. Disentanglement is achieved by jointly decomposing all such operators across the dataset via a common change of basis. This joint diagonalization leads to irreducible blocks that may correspond to distinct factors of variation, as illustrated in Fig. 7b of the original MSP paper. The subsequent NFT work further links this structure to disentanglement via Fourier frequencies.
>
> By contrast, **SKD** (and its extension, **SSM-SKD**) is grounded in Koopman operator theory and emphasizes spectral decomposition. In SKD, static factors of variation are associated with Koopman eigenvalue 1, and dynamic factors with other eigenvalues. This perspective directly ties disentanglement to the spectral properties of the learned dynamics. Practically, SKD performs eigendecomposition *per batch* rather than jointly across the dataset, and extracts factors directly from the Koopman operator’s eigenstructure. SSM-SKD further simplifies this process by performing eigendecomposition *per sequence* and constraining the eigenbasis to contain only a single eigenvalue 1 and leveraging projections into and out of the Koopman eigenbasis. This approach avoids the need for additional sequences or dataset-wide operator alignment, focusing entirely on the intrinsic structure of each sequence’s dynamics.
>
> While both MSP and SKD rely on latent linear operators, they differ meaningfully in **motivation**, **decomposition approach**, and **practical implementation**. We note that SKD and SSM-SKD are also evaluated across a broad range of disentanglement benchmarks and data modalities, whereas MSP’s applicability to disentanglement has thus far only been investigated on a limited set of examples.
>
> In response to your suggestion, we will revise the related work section to include a discussion of MSP and its follow-up works, emphasizing their relevance and contributions to disentanglement. Additionally, we have begun integrating MSP into our benchmark framework and, provided that we manage to do so successfully, will report its results as soon as they become available.

---

> > ### Comment · Reviewer_53Qu · 2025-08-04
> >
> > I appreciate the author's detailed response! I will raise my score by one now that my concerns have been clarified.

---

> > ### Author Response · Authors · 2025-08-06
> > **Thank You**
> >
> > Thank you very much for engaging in the discussion. We greatly appreciate your valuable and constructive feedback.

---

### Official Review · Reviewer_sukC · 2025-07-07

**Rating:** 4
**Confidence:** 3

**Summary:**

This paper proposes the Multi-factor Sequential Disentanglement (MSD) Benchmark, designed for evaluating multi-factor sequential representation learning that goes beyond the conventional static–dynamic factor dichotomy. The authors make four key contributions:

* Six cross-modal datasets (covering video, audio, and time series) with a unified format, along with publicly available generation scripts and factor labels, addressing the data fragmentation issue in prior work.

* Ten evaluation metrics, including multi-factor extensions of existing swap/sample measures and three video consistency metrics.

* Latent Exploration Stage (LES), which automatically maps semantic factors to latent variables via predictors or swap-based interventions, significantly improving the evaluation efficiency and scores of existing models.

* Single-Static-Mode SKD (SSM-SKD), which achieves state-of-the-art performance on four datasets (with an ≈16% improvement on Sprites).

Additionally, the authors leverage Vision-Language Models (VLMs) as automated taggers and judges for annotation and zero-shot evaluation, demonstrating high consistency with human assessments.

**Dataset Code Accessibility:**

Yes

**Ethical Considerations:**

No, there are no or only very minor ethics concerns

**Final Justification:**

After discussion and considering the suggestions of other reviewers, I maintain my score of 4.

**Limitations Weaknesses:**

* Dependence on GPT-4o: The VLM Judge relies on commercial APIs. Future changes in model policies or pricing could impact reproducibility and fairness of evaluations.

* Synthetic or weakly labeled data: Although the benchmark covers multiple modalities, the majority of the video data is still synthetic animation; real-world videos are only used for qualitative analysis or automated tagging.

* Metric aggregation decisions: The current approach of aggregating multiple metrics via weighted averaging may mask certain failure modes. It is recommended to disclose the aggregation weights and provide a per-metric leaderboard for greater transparency.

**Strengths Contributions:**

The problem is important and novel: Multi-factor sequential disentanglement is essential for real-world applications (e.g., facial expressions, music, air quality) but has long been overlooked due to the dominance of the static–dynamic binary factorization.

* LES establishes factor–latent correspondence with minimal human effort and is more stable than manual matching (see Table 5).

* SSM-SKD cleverly compresses all static information into a single feature vector via a sample-level Koopman operator, achieving strong performance.

* Cross-modal coverage: The inclusion of audio (dMelodies-WAV) and real-world air quality data further broadens the benchmark’s applicability.

---

> ### Author Rebuttal · Authors · 2025-07-30
>
> We thank the reviewer for taking the time to provide a thoughtful and constructive review. We also appreciate the recognition of the importance of the problem, the novelty of our proposed procedures, the diversity of our benchmarking, and the overall contribution of our work. Below, we address each of the reviewer’s concerns. Given the opportunity, we would be happy to incorporate the suggested changes in the final revision and provide further clarification if needed.
>
> ---
>
> > ### **Comment:**
> > *Dependence on GPT-4o: The VLM Judge relies on commercial APIs. Future changes in model policies or pricing could impact reproducibility and fairness of evaluations.*
>
> ### **Response:**
> We fully agree that long-term reproducibility and fairness require accessible, non-proprietary tools. Motivated by this feedback, we have added support for the open-source VLM **Qwen2.5** as a default option in our benchmark. The integration required minimal changes thanks to our modular design, and pretrained weights are fetched seamlessly from the Hugging Face Hub. While GPT-4o was originally chosen for its easy integration and popularity, the benchmark was built from the outset to support interchangeable VLM backends. We are committed to expanding this support further in future releases to promote broader adoption and long-term sustainability.
>
> ---
>
> > ### **Comment:**
> > *Synthetic or weakly labeled data: Although the benchmark covers multiple modalities, the majority of the video data is still synthetic animation; real-world videos are only used for qualitative analysis or automated tagging.*
>
> ### **Response:**
> We would like to emphasize that, as demonstrated in Section 3.4, all existing methods currently fail on real-world datasets. As a result, quantitative evaluation in such cases may offer limited insights. However, in Section 4.4, we show that by leveraging vision-language models (VLMs), it is indeed possible to benchmark and quantitatively assess performance on real-world datasets. This indicates that the primary bottleneck lies in the methods themselves, rather than the datasets. If a model were capable of effectively handling real-world datasets such as CelebA, our benchmarking framework would readily support quantitative evaluation. Therefore, the development of new methods for real-world data is feasible within our benchmark, and we hope this will encourage further progress in multifactor disentanglement research.
>
> ---
>
> > ### **Comment:**
> > *Metric aggregation decisions: The current approach of aggregating multiple metrics via weighted averaging may mask certain failure modes. It is recommended to disclose the aggregation weights and provide a per-metric leaderboard for greater transparency.*
>
> ### **Response:**
> We thank the reviewer for this insightful suggestion. We agree that aggregating metrics into a single score may obscure individual failure modes, and we have updated our paper accordingly to improve transparency and interpretability.
>
> First, we clarify how the scores in Table 1 (in the main paper) are computed. For each model *m* on dataset *d*, we calculate the unweighted average of all applicable normalized metric scores:
>
> $$S_{m,d} = \frac{1}{K_d} \sum_{k=1}^{K_d} s_{m,d,k}$$
>
> Where:
> - $S_{m,d} \in [0, 1]$ is the aggregated score for model *m* on dataset $d$,
> - $s_{m,d,k}$ is the normalized score on metric $k$,
> - $K_d$  is the number of applicable metrics for dataset $d$.
>
> To avoid masking metric-specific behaviors, and inspired by the reviewer’s suggestion, we have added a new **per-metric leaderboard table** that presents each model’s performance per metric, averaged across all datasets where that metric is valid:
>
> $$L_{m,k} = \frac{1}{|D_k|} \sum_{d \in D_k} s_{m,d,k}$$
>
> Where:
> - $L_{m,k}$ is the leaderboard score of model $m$ on metric $k$,
> - $\mathcal{D}_k$ is the set of datasets for which metric $k$ is applicable.
>
> Each **column** corresponds to a metric, and each **row** represents a rank level. Row 1 contains the top-performing model(s) for each metric, Row 2 the second-best, and so on. This layout makes it easy to observe which models perform well under specific evaluation criteria and complements the aggregated view in Table 1 of the main paper.
>
> In addition, we include:
> - **Raw scores** for all *(model, dataset, metric)* combinations in the supplementary material;
> - **Tables 8–13 in Appendix G**, detailing per-metric results across datasets.
>
> > Leaderboard showing the mean and standard deviation of each model's performance across metrics. Higher values indicate better performance.
>
> | Rank | M-Swap ↑      | M-GSample ↑   | DCI-M ↑      | DCI-C ↑      | DCI-E ↑      | C-Swap ↑     | C-Sample ↑   | GC-Sample ↑  | 2-Swap ↑     | 2-GSample ↑  |
> |------|----------------|----------------|---------------|---------------|---------------|---------------|---------------|----------------|----------------|----------------|
> | 1    | SSM-SKD (0.79 ± 0.14) | SSM-SKD (0.80 ± 0.14) | SSM-SKD (0.49 ± 0.39) | SSM-SKD (0.81 ± 0.14) | SSM-SKD (0.78 ± 0.24) | SSM-SKD (0.83 ± 0.14) | SSM-SKD (0.95 ± 0.02) | SSM-SKD (0.96 ± 0.02) | SKD (0.87 ± 0.12) | SKD (0.87 ± 0.11) |
> | 2    | SKD (0.69 ± 0.06) | SKD (0.70 ± 0.06) | VAE (0.27 ± 0.32) | β-VAE (0.79 ± 0.04) | Sparse-AE (0.55 ± 0.31) | MGP-VAE (0.66 ± 0.26) | β-VAE (0.95 ± 0.05) | β-VAE (0.95 ± 0.06) | SSM-SKD (0.86 ± 0.14) | SSM-SKD (0.86 ± 0.13) |
> | 3    | MGP-VAE (0.68 ± 0.14) | VAE (0.67 ± 0.11) | β-VAE (0.26 ± 0.15) | VAE (0.78 ± 0.10) | β-VAE (0.53 ± 0.24) | Sparse-AE (0.56 ± 0.27) | SKD (0.94 ± 0.04) | SKD (0.95 ± 0.03) | MGP-VAE (0.79 ± 0.09) | β-VAE (0.74 ± 0.14) |
> | 4    | VAE (0.67 ± 0.11) | β-VAE (0.67 ± 0.03) | Sparse-AE (0.25 ± 0.29) | Sparse-AE (0.77 ± 0.09) | SKD (0.53 ± 0.17) | SKD (0.53 ± 0.07) | Sparse-AE (0.91 ± 0.11) | Sparse-AE (0.93 ± 0.08) | β-VAE (0.73 ± 0.14) | VAE (0.70 ± 0.15) |
> | 5    | Sparse-AE (0.67 ± 0.07) | Sparse-AE (0.64 ± 0.04) | MGP-VAE (0.17 ± 0.22) | SKD (0.62 ± 0.06) | VAE (0.47 ± 0.28) | β-VAE (0.49 ± 0.22) | VAE (0.90 ± 0.06) | VAE (0.93 ± 0.06) | VAE (0.70 ± 0.15) | MGP-VAE (0.69 ± 0.09) |
> | 6    | β-VAE (0.66 ± 0.03) | MGP-VAE (0.57 ± 0.06) | SKD (0.13 ± 0.11) | MGP-VAE (0.47 ± 0.14) | MGP-VAE (0.42 ± 0.27) | VAE (0.43 ± 0.35) | MGP-VAE (0.77 ± 0.04) | MGP-VAE (0.74 ± 0.04) | Sparse-AE (0.70 ± 0.12) | Sparse-AE (0.66 ± 0.06) |

---

### Note · Authors · 2025-08-13

We thank the reviewers for their constructive feedback and recognition of our benchmark’s novelty and potential impact. Below we summarize the main concerns and the changes we made in response.

- **Reproducibility with proprietary VLMs (sukC)** - To address concerns about relying solely on GPT-4o, we integrated the open-source **Qwen2.5** VLM as a default backend. Thanks to our modular design, VLMs can be swapped seamlessly, ensuring long-term accessibility and fairness.

- **Use of real-world datasets (sukC)** - Reviewers noted that quantitative evaluation relied mainly on synthetic video. We showed (Sec. 4.4) that VLM-based evaluation enables quantitative benchmarking in real-world datasets such as CelebA. While all current methods underperform in these domains, our framework supports evaluating future methods that succeed, encouraging progress beyond synthetic settings.

- **Metric transparency (sukC, NcTT)** - We clarified aggregation with explicit equations and now provide: (i) an unweighted averaging scheme for Table 1, (ii) a new **per-metric leaderboard** to expose strengths and weaknesses, and (iii) raw results and per-metric tables in the appendix.

- **Metric definitions and configurations (NcTT, KVH3)** - We expanded Sec. 3.4 and Appendix D with a structured taxonomy of all ten metrics (Factor Swap/Sample, DCI, Consistency), detailed configurations, and qualitative examples. DCI metrics were adapted to sequential multi-factor settings with full implementation details for reproducibility.

- **Relationship between SSM-SKD and MSP (53Qu)** - We added a detailed comparison in the related work section, clarifying theoretical and practical differences, and began integrating MSP into our benchmark for future evaluation.

- **Additional checkpoints for reproducibility (NcTT)** - We trained **522 new model checkpoints** across datasets, covering impactful hyperparameter settings. These will be released publicly after the review period.

- **Clarity and presentation (KVH3)** - We refined notation, clarified latent output requirements, and enhanced metric explanations to improve accessibility across modalities.

These revisions strengthen the benchmark’s transparency, reproducibility, and applicability, directly addressing all major reviewer concerns.

---

### Decision · Program_Chairs · 2025-09-18

**Decision:**

Accept (poster)

**Comment:**

This paper introduces a benchmark for multi-factor sequential disentanglement, spanning vision, audio, and time series. It combines reformatted datasets, carefully designed metrics, and automated evaluation via VLMs. Concerns regarding metrics, real-world data reliance, and VLM reproducibility were addressed in rebuttals and revisions. Importantly, the paper sets a new standard for disentanglement research by embracing multi-factor sequential settings and providing the necessary infrastructure to drive the field forward. Given its novelty, comprehensiveness, and potential for community impact, I recommend acceptance.